# Paralog dependency indirectly affects the robustness of human cells

Rohan Dandage[1,2,3,4,5] ID & Christian R Landry[1,2,3,4,5,*] ID

## Abstract

The protective redundancy of paralogous genes partly relies on the fact that they carry their functions independently. However, a significant fraction of paralogous proteins may form functionally dependent pairs, for instance, through heteromerization. As a consequence, one could expect these heteromeric paralogs to be less protective against deleterious mutations. To test this hypothesis, we examined the robustness landscape of gene loss-of-function by CRISPR-Cas9 in more than 450 human cell lines. This landscape shows regions of greater deleteriousness to gene inactivation as a function of key paralog properties. Heteromeric paralogs are more likely to occupy such regions owing to their high expression and large number of protein–protein interaction partners. Further investigation revealed that heteromers may also be under stricter dosage balance, which may also contribute to the higher deleteriousness upon gene inactivation. Finally, we suggest that physical dependency may contribute to the deleteriousness upon loss-of-function as revealed by the correlation between the strength of interactions between paralogs and their higher deleteriousness upon loss of function.

**Keywords** CRISPR; gene dosage; gene duplication; paralogs; protein–protein interactions
**Subject Categories** Computational Biology; Evolution & Ecology
**Mol Syst Biol.** (2019) 15: e8871

## Introduction

After a gene duplication event and before they become functionally distinct, paralogs are redundant and can mask each other's inactivating mutations (Pickett & Meeks-Wagner, 1995; Brookfield, 1997; Diss *et al*, 2014). This mutational robustness does not provide an advantage strong enough by itself to cause the maintenance of paralogs by natural selection unless mutation rate or population size is exceptionally large (van Nimwegen *et al*, 1999). Nevertheless,

paralogous genes affect how biological systems globally respond to loss-of-function (LOF) mutations. For instance, the early analysis of growth rate of the yeast gene deletion collection revealed that genes with duplicates are enriched among the ones that have a weak effect on fitness when deleted (Gu *et al*, 2003). Likewise, singletons (genes with no detectable homologous sequence in the genome) tend to be overrepresented among genes whose deletion is lethal. Further studies in yeast also showed that redundancy could be maintained for millions of years, making the impact of duplication long lasting (Dean *et al*, 2008). A parallel observation in humans showed that genes are less likely to be involved in diseases if they have a paralog, and the probability of disease association for a gene decreases with increasing sequence similarity with its closest homolog in the genome (Hsiao & Vitkup, 2008). These observations, along with smaller scale observations made in classical genetics (Pickett & Meeks-Wagner, 1995; Diss *et al*, 2014), strongly demonstrate that redundancy allows paralogs to compensate for each other's LOF at the molecular level.

The buffering ability of paralogs is however not universal (Ihmels *et al*, 2007), and opposite results have been reported. For instance, Chen *et al* (2013b) reported an enrichment of human diseases among paralogous genes, particularly among the ones with higher functional similarity. The authors explained this result with a model in which redundancy reduces the efficacy of purifying selection, leading to the maintenance of disease alleles that could have lower penetrance, for instance, through noise in gene expression. Other authors have shown that the retention of whole-genome duplicates could be biased toward genes that are more likely to bear autosomal-dominant deleterious mutations (Singh *et al*, 2012). In this case, the maintenance of paralogs would be associated with greater susceptibility to disease mutations, contrary to the robustness expected from gene redundancy. A better understanding of whether and how paralogs can compensate for each other's deleterious mutations therefore requires a better understanding of the mechanisms involved. This would improve our understanding of evolution and also accelerate the development of medical interventions because redundancy is often a major obstacle in this context (Lavi, 2015).

The mechanisms by which paralogs compensate for each other's LOF mutations are for most cases not known in details (Pickett &

1   Département de Biologie, Université Laval, Québec, QC, Canada
2   Département de Biochimie, Microbiologie et Bio-Informatique, Université Laval, Québec, QC, Canada
3   Institut de Biologie Intégrative et des Systèmes (IBIS), Université Laval, Québec, QC, Canada
4   The Québec Network for Research on Protein Function, Engineering, and Applications (PROTEO), Université Laval, Québec, QC, Canada
5   Centre de Recherche en Données Massives (CRDM), Université Laval, Québec, QC, Canada
    *Corresponding author. Tel: +1 418 656 3954; E-mail: christian.landry@bio.ulaval.ca

Meeks-Wagner, 1995; Diss *et al*, 2014), but likely involve active and passive mechanisms, from transcriptional to post-translational ones. For instance, it was shown for a small fraction of paralogous gene pairs that a member of a pair is upregulated by some feedback mechanism upon the deletion of the second copy (Kafri *et al*, 2005). Although it may have important consequences, the occurrence of this phenomenon is however very likely limited. Indeed, a systematic assessment of this mechanism at the protein level in yeast found that it could take place only for a very small set of paralogous genes (DeLuna *et al*, 2010).

Another potential mechanism of compensation takes place at the level of protein–protein interactions (PPI) (reviewed by Diss *et al*, 2014), whereby paralogs replace each other with respect to their binding partners through ancestrally preserved binding ability. Evidence for this mechanism was recently reported by Diss *et al* (2013, 2017). The model proposed is that paralogs appear to have different binding partners in wild-type cells, because they mutually exclude each other from binding with potential partners. This is due to differential binding affinity or expression levels of the paralogs that tilts binding competition toward one paralog or the other. Upon deletion, the mutual exclusion is relieved and compensation becomes apparent. Results consistent with this observation were obtained by Ori *et al* (2016) in mammalian cells. The authors showed that some paralogs can replace each other through changes in expression within protein complexes, supporting the fact that paralogs have preserved the ability to interact with the same partners. Another study reported observations consistent with this model using proteomics analyses of cancer cell lines (Gonçalves *et al*, 2017). In this case, an increased copy number for one gene led to increased protein abundance and a decrease in abundance of its paralogs, as if a feedback mechanism was affecting the balance between paralogs. This feedback is likely due to post-translational regulation that leads to the degradation of the displaced paralogs from protein complexes, also called protein attenuation (Ishikawa *et al*, 2017; Taggart & Li, 2018). This observation suggests that the two paralogs would have overlapping binding partners and the balance would be determined by their relative affinity and abundance, as observed in one recent meta-analysis study (Sousa *et al*, 2019). Finally, Rajoo *et al* (2018) examined the composition of the yeast nuclear pore complex and, similarly to the Diss *et al* study (Diss *et al*, 2013), found that paralogous proteins can at least partially replace each other *in situ* upon deletion and change in abundance.

A major determinant that limits the ability of paralogs to compensate is their functional divergence, which can be approximated by sequence divergence (Hsiao & Vitkup, 2008; Li *et al*, 2010). Other factors could also play a role, for instance, cross-dependency, which has been brought to light only recently. DeLuna *et al* (2010) looked at protein abundance of yeast paralogs when their sister copies are deleted, and found that six of the 29 pairs studied displayed negative responsiveness: Upon deletion, the remaining paralog showed a decreased protein abundance. In half of these cases, the paralogs heteromerized (physically interacted with each other), suggesting that protein abundance may depend on their physical interactions. The control of protein abundance through interactions was also recently elucidated in the context of human cells (Sousa *et al*, 2019). The consequences of these decreases in abundance were not investigated further but one could

imagine that this would directly affect the compensating ability of paralogs, because the deletion of one copy of a pair leads to a LOF of the second, thereby essentially acting as a dominant negative effect. A recent study by Diss *et al* (2017) directly examined paralog compensation at the level of protein–protein interactions. Among more than 50 pairs of paralogs, they showed that not all paralogs could compensate in the yeast protein interaction network. About 20 pairs showed dependency, i.e., one paralog lost some or all its interaction partners upon the loss of the second. Diss *et al* found that dependent pairs were enriched for pairs that form heteromers and, in some cases, the dependency could be explained by a strong decrease in protein abundance upon deletion, consistent with the observation of DeLuna *et al* (2010).

Altogether, these observations raise the possibility that heteromerization of paralogs may reflect their physical and functional dependency, which as a consequence would reduce the ability of paralogous genes to compensate for each other's loss. One could therefore predict that the protection that paralogous genes provide against the effect of LOF mutations would be contingent on whether their products form heteromeric complexes with each other or not. These genes would have fitness effects that are closer to that of single copy genes (singletons) than that of typical duplicates. Here, we examine these predictions by re-analyzing a set of well-curated pairs of human paralogous genes (Singh *et al*, 2015; Lan & Pritchard, 2016) and recent large-scale genome-wide CRISPR-Cas9 screens in which the effect of gene LOF on cell proliferation was examined in more than 450 cancer cell lines (Wang *et al*, 2015; DepMap, 2018) and a primary cell line (Shifrut *et al*, 2018). The meta-analysis of the effect of gene LOF on cell proliferation, mRNA expression from 374 cell lines, protein expression from 49 cell lines and protein–protein interactions (Table EV1) revealed patterns which strongly support our hypothesis that paralogs that assemble are less protective, but through factors other than heteromerization itself.

## Results

### Paralogous genes protect against the effect of gene LOF across all cell lines

We used two datasets of paralogous genes, one of relatively young paralogs, largely derived from small-scale duplications (Lan & Pritchard, 2016) and another set of relatively old paralogs most likely derived from whole-genome duplication (Data ref: Ohnolog, 2018; total of 3,132 pairs of paralogs, see Materials and Methods, Dataset EV1). We first examined whether paralogous genes protect against the deleterious effects of LOF mutations in a set of 455 human cell lines from three independent CRISPR-Cas9 genome-wide LOF screens (Table EV1). Such experiments yield a CRISPR score (CS) per gene which is an estimate of the relative depletion of guide RNAs (gRNAs) during the genome-wide CRISPR-Cas9 screening experiment. CS therefore reflects the relative deleteriousness of LOF on cell proliferation (Fig EV1): A lower CS value indicates more deleteriousness and vice versa. These datasets are (i) CS1 from four cell lines (Wang *et al*, 2015), (ii) CS2 from 450 cell lines (Meyers *et al*, 2017; DepMap, 2018), (iii) CS2.1 from 450 cell lines (DepMap, 2018), and (iv) CS3 from 1 primary cell line (Shifrut *et al*, 2018; see

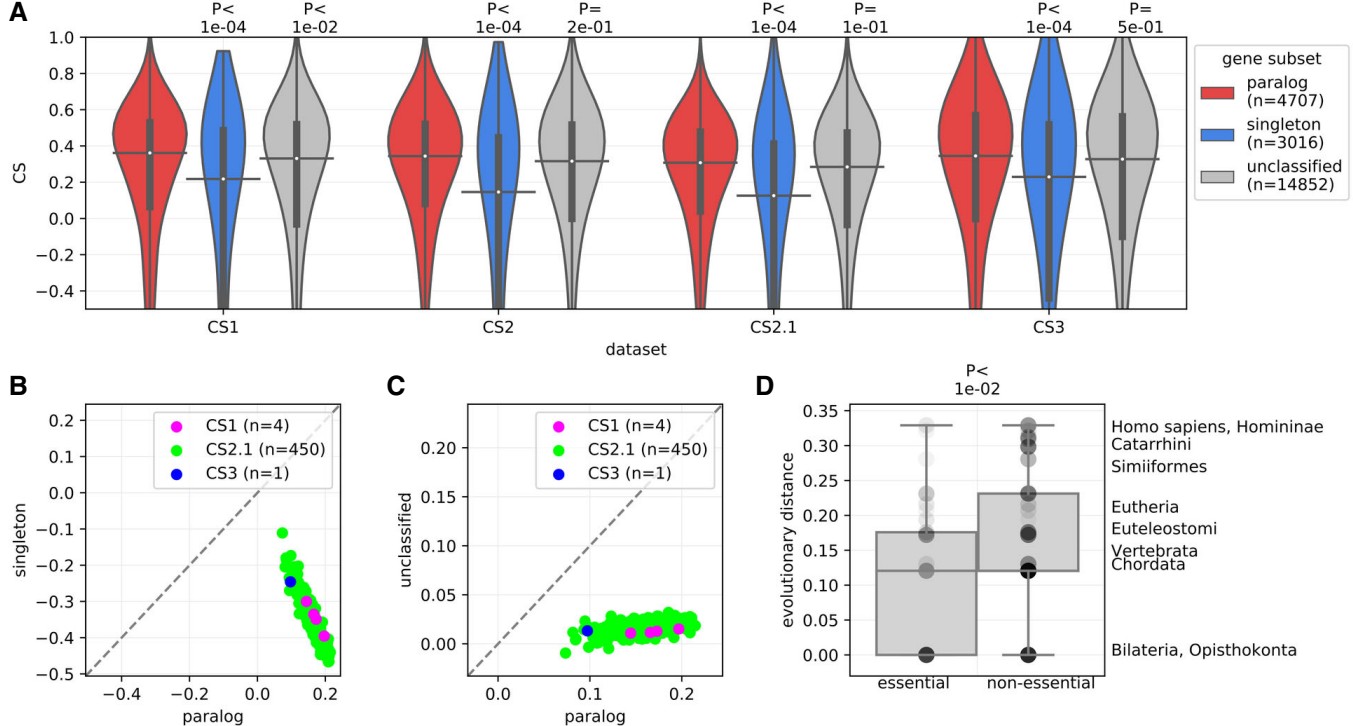

**Figure 1. The LOF of paralogs is less deleterious than that of singletons in human cell lines.**

A   LOF data derived from genome-wide CRISPR-Cas9 screening experiments. The deleteriousness of LOF of a gene on cell proliferation is estimated from the depletion of gRNAs in the experiment. The extent of depletion is measured as a CRISPR score (CS, see Materials and Methods). CS values across cell lines from three biologically independent datasets—CS1 (Wang *et al*, 2015), CS2/CS2.1 (Meyers *et al*, 2017; DepMap, 2018), and CS3 (Shifrut *et al*, 2018) are shown. Genes that are not in the paralog datasets but that were not identified as singletons in the stringent identification of singletons are denoted as "unclassified". Relatively higher CS of paralogs compared to singletons indicates that they are relatively less deleteriousness. *P*-values from two-sided Mann–Whitney *U* tests are shown. On the violin plots, the medians of the distributions are shown by a horizontal black line and quartiles by a vertical thick black line. For clarity, the upper and lower tails of the distributions are not shown.

B, C   (B) Comparisons of CS values between paralogs and singletons and (C) between paralogs and unclassified genes (neither clearly a paralog nor a singleton). CS data for 4 (CS1) + 450 (CS2.1) + 1 (CS3) cell lines is shown. Each point represents the mean CS for a class (singleton, paralog, or unclassified) in an individual cell line. All points are below the diagonal (dashed gray line), showing that the effect is systematic and largely cell-line independent. Similar plots are shown for CS2 dataset in Appendix Fig S2.

D   Older paralogs tend to be more essential than younger ones and are therefore less protective (i.e., more deleterious upon LOF). On the *y*-axis, the age groups are ordered in increasing distance of phylogenetic node of duplication relative to common ancestor, i.e. Opisthokonta. Sets of essential and non-essential genes were derived from the union of gene sets reported by DepMap (2018) and BAGEL (Hart & Moffat, 2016; See Materials and Methods). *P*-value from a two-sided Mann–Whitney *U* test is shown. The boxes represent the first and third quartiles (Q1 and Q2) of the distribution, and the upper and lower whiskers extend up to Q3 + 1.5*interquartile range and Q1 − 1.5*interquartile range, respectively. The central horizontal line represents the median of the distributions containing 65 data points in the case of essential paralogs and 235 data points in the case of non-essential paralogs.

Source data are available online for this figure.

Dataset EV2 for cell-line information, Dataset EV3 for gene-wise CS values). All the CS values capture the essentiality of the genes which, in the case of cancer cell lines, are found to be largely independent of the role of the genes in cancerogenesis (Fig EV1). Because the estimation of CS of the paralogs could be confounded by gRNAs that match to more than one gene due to their sequence similarities, we recomputed scores for the CS1, CS2.1, and CS3 datasets by considering only the gRNAs that uniquely align to the genome (see Materials and Methods). Dataset CS2 and dataset CS2.1 constitute data from the same set of cell lines (biologically identical), but analyzed differently. CS2 takes copy-number variation effects in each cell line into account (used directly as computed by the authors; Meyers *et al*, 2017), while CS2.1 is analyzed by utilizing only the uniquely aligned gRNAs (see

Materials and Methods). CS values among datasets CS1 and CS2/CS2.1 are well correlated, indicating reproducible measurements of fitness effects across platforms, methodologies, cell lines, and cell types (Appendix Fig S1). The weaker correlation with dataset CS3 values (Spearman correlation coefficient ranges from 0.19 to 0.21), however, could be attributed to the difference in the physiology of the primary and cancer cell lines itself, although technical factors could also be responsible.

As expected, we find that paralogs buffer the effect of gene LOF. Genes with paralogs have relatively higher CS values than singletons (see Materials and Methods for classification of singletons), for the three biologically independent datasets considered (Fig 1A). To confirm that these effects were systematic and were not driven by few cases of cell lines with strong effects, we compared the mean

CS for paralogs and singletons across cell lines (see Fig 1B for analysis with CS2.1 and Appendix Fig S2A for analysis with CS2 dataset). All cell lines systematically showed stronger buffering effects for the inactivation of paralogs compared to singletons, with no exception. The same results were observed for the comparison of paralogs with genes that are not in the set of paralogs nor classified as singletons, denoted as "unclassified" (see Fig 1C for analysis with CS2.1 and Appendix Fig S2B for CS2 dataset). These results are therefore highly reproducible and cell-line independent. However, the trend showed some dependence on molecular features such as mRNA expression levels, as we discuss below.

### Older paralogs tend to be less protective

In order to determine the effect of paralog age on deleteriousness, we compared the essential and non-essential sets of genes in terms of their age group of duplications retrieved from Ensembl Compara (Herrero *et al*, 2016; see Materials and Methods). We find that, albeit with a weak difference, older paralogs are more likely to be classified as essential genes and thus have potentially more deleterious effects upon LOF than younger paralogs (Fig 1D, see Materials and Methods for the classification of essential genes). This result underscores similar findings from earlier studies showing that the more diverged paralogs are, the less likely they are to buffer each other's loss, in the context of human diseases or yeast gene deletions (Hsiao & Vitkup, 2008; Li *et al*, 2010; Plata & Vitkup, 2014).

### Heteromeric paralogs emerge from ancestral homomers

The model in which paralogous genes are dependent on each other considers that interacting paralogs derive from ancestral homomeric proteins (Bridgham *et al*, 2008; Baker *et al*, 2013; Kaltenegger & Ober, 2015; Diss *et al*, 2017). We can assume that when the two paralogs individually form a homomer, the ancestral protein was most likely also a homomer. Therefore, we can infer that heteromers of paralogs are derived from ancestral homomers, if each paralog also forms a homomer. Homomeric, in the context of this study, refers to the assembly of a protein with itself while heteromers of paralogs or heteromeric paralogs refer to paralogous proteins that assemble with each other.

We used two sources of PPI, BioGRID (Chatr-Aryamontri *et al*, 2015, 2017) and IntAct (Orchard *et al*, 2014), to define homomeric genes or heteromeric gene pairs based on PPI (see Materials and Methods). Further, the subsets were defined based on all PPI (henceforth, this dataset will be referred to as "all PPI") or direct physical interactions only (henceforth, this dataset will be referred to as "direct PPI"). Considering all PPIs (see Materials and Methods for the difference between "all PPI" and "direct PPI"), paralogs are 8.13 times more likely to form heteromeric pairs (Fisher's exact test, *P*-value < 1.4e-14) if they also both form homomers than if none of them does. The likelihood is 48.88 times for heteromers defined by "direct PPI"s only (*P*-value < 5.5e-18; see Appendix Table S1 for the numbers of pairs in each category). We can therefore generally assume that pairs of heteromeric paralogs are more likely to be derived from ancestral homomers, consistent with previous observations (Wagner, 2003; Pereira-Leal *et al*, 2007).

### Paralogs that form heteromers have stronger effects on cell proliferation when inactivated

Next, we investigated the effect of LOF of paralogs that form heteromers and those that do not. Consistent with the dependency hypothesis, the LOF of heteromeric paralogs seems to cause relatively more deleterious effect on cell proliferation than the LOF of non-heteromeric paralogs, across all 4 CS datasets (Fig 2A, similar analysis with "direct PPI"s is shown in Appendix Fig S3). We also observe that the effect is consistent across cell lines by looking at the mean CS of heteromers or non-heteromers within each cell line (Fig 2B), with a majority of cell lines showing stronger effects for the LOF of paralogs forming heteromers. This trend is clearly observed across all the CS datasets and irrespective of the source of the PPI used for the definition of the heteromeric paralogs (similar analysis as that of the Fig 2B with all the rest of the combinations of the PPI sources and CS datasets is shown in Appendix Fig S4). A similar analysis with paralogs that are both heteromers and homomers compared with paralogs which are only homomers shows that interacting paralogous are relatively more deleterious (Fig 2C). This trend is also clearly observed across all the CS datasets and irrespective of the source of the PPI used for the definition of the subsets of paralogs (similar analysis as that of the Fig 2B with all the rest of the combinations of the PPI sources and CS datasets is shown in Appendix Fig S4). The effects are therefore not due to homomerization but due to heteromerization (Fig 2C). These results support the hypothesis that interacting pairs of paralogs are less likely to buffer each other's LOF.

One potential confounding factor with this analysis is the fact that the frequency of heteromers could covary with the age of paralogs, which we showed above to affect at least partially the essentiality of the gene (Fig 1D). Heteromers are indeed older than the non-heteromeric paralogs (Fig 2D), albeit only in the case of the heteromers defined by "all PPI"s. We therefore looked at CS values of paralog LOF corrected for age, by using age groups. We observed that for all age groups, except for two, CS values for heteromers are indeed lower than for non-heteromers, suggesting that this effect is largely independent from age (Fig 2E). The reason for the inconsistency between the two age groups is however unclear, the potential confounding factors could be the DNA sequence divergence between paralogs and the ability of gRNA to target one gene specifically.

### Molecular functions enriched for heteromeric paralogs tend to be more critical for cell proliferation

It is possible that the effects detected are due to specific gene functions that would be particularly associated with heteromeric paralogs. We first examined whether heteromers of paralogs are enriched for particular function among all paralogs. We found that heteromers of paralogs are enriched for gene sets containing proteins that have catalytic activity and known to directly interact/regulate with each other such as kinase binding (Breitkreutz *et al*, 2010) as well as DNA binding proteins from the histone deacetylase binding gene set (see Dataset EV4 gene sets and GO terms used in the analysis, Dataset EV5 for enrichment analysis).

From this gene set enrichment analysis, we find that the proportion of heteromerization of paralogs in a gene set, in general, is negatively correlated (Spearman correlation coefficient = −0.26, *P*-value = 0.086) with the average CS value of paralogs per gene set.

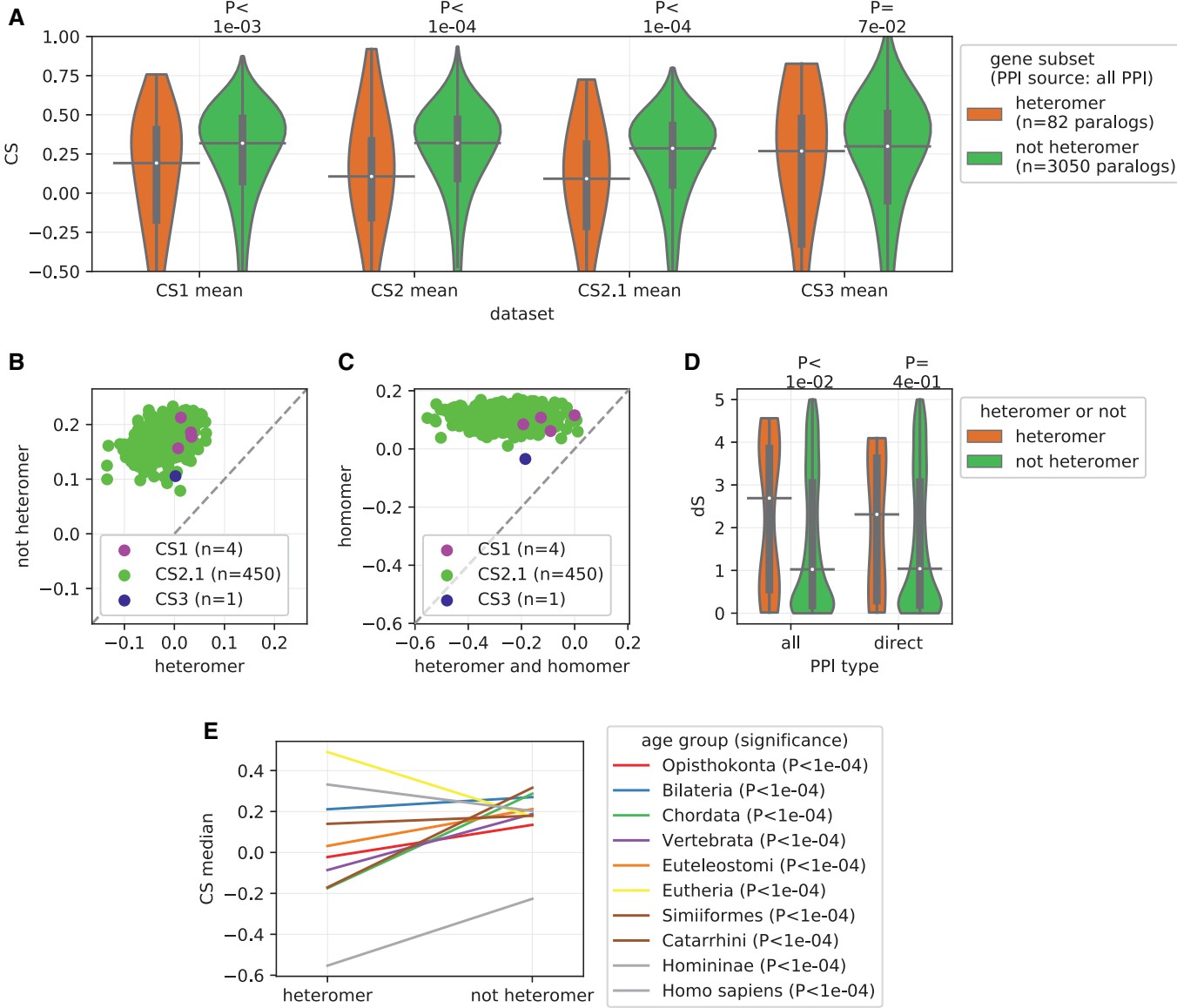

**Figure 2. The LOF of paralogs that form heteromers is more deleterious than the LOF of non-heteromers.**

A The effect of LOF on cell proliferation (CS values) is relatively more deleterious in the case of heteromeric paralogs than non-heteromers, across all 4 CS datasets. *P*-values from two-sided Mann–Whitney *U* tests are shown. Similar plot for heteromers defined with direct PPI only is shown in Appendix Fig S3.

B Mean CS values of heteromeric paralogs and non-heteromers (defined by "all PPI"s from BioGRID source) are shown across cell lines. Each point represents the mean CS value for a class in an individual cell line. All the points are above the diagonal (dashed gray line), showing that the effect is systematic and largely independent of cell line. Similar plots for both PPI sources and CS2 dataset are shown in Appendix Fig S4.

C Similar to panel (B), but comparing paralogs that form heteromers and homomers to those that form homomers only (defined by "all PPI"s from BioGRID source). This result shows that the difference between heteromers and non-heteromers is not caused by the fact that heteromers are also enriched for homomers. Similar plots for both PPI sources and CS2 dataset are shown in Appendix Fig S4.

D Paralogs that form heteromers tend to have been duplicated earlier in evolution. The age of the paralog pairs is shown in terms of synonymous substitutions per site (dS) (see Materials and Methods), a proxy for age. Data are shown for interactions derived from "all PPI", and those that are more likely to detect "direct PPI". *P*-values from two-sided Mann–Whitney *U* tests are shown.

E Paralogs that form heteromers tend to be more deleterious upon LOF than other paralogs. Data from CS2.1 are shown, largely independent of the age of the paralog. In the legends, paralogs are ordered by their age. The CS values per class of paralogs (heteromer or not) and their age group are aggregated by taking median across cell lines. Note that while heteromers are more deleterious in most of the age groups, in the case of 2 out of 10 age groups a reverse trend is observed. Distributions of the CS values per class of paralogs (heteromer or not) and their age group for this analysis are shown in Appendix Fig S5A. Similar analysis with dataset CS2 and for heteromers detected with "direct PPI"s only is shown in Appendix Fig S5 B–D. *P*-values from two-sided Mann–Whitney *U* tests are shown.

Data information: On the violin plots (panel A and D), the medians of the distributions are denoted by a horizontal black line, while the quartiles of the distributions from the median value are indicated by a vertical thick black line. For clarity, the upper and lower tails of the distributions are not shown in panel (A).
Source data are available online for this figure.

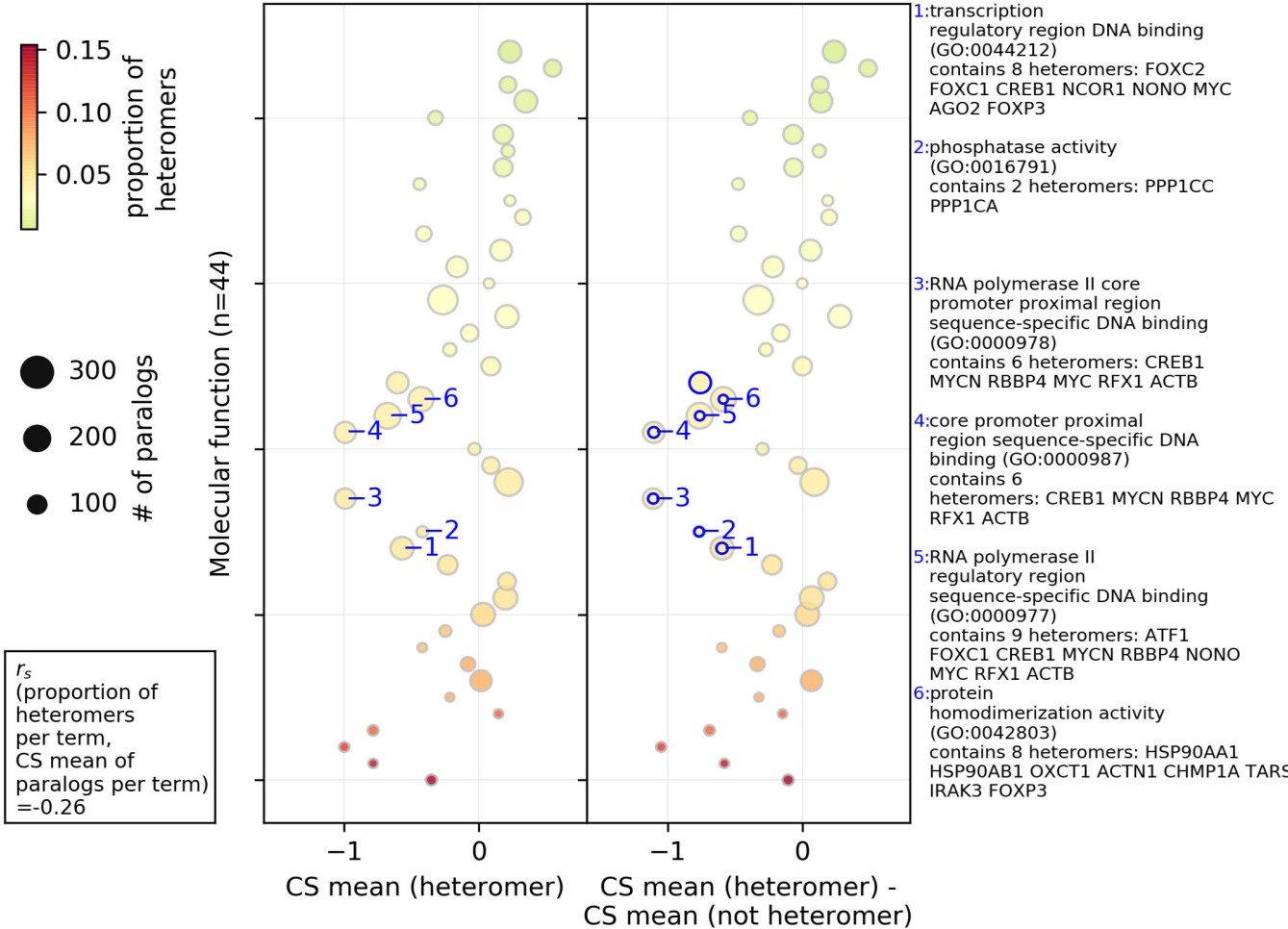

**Figure 3. Association between the molecular functions of paralogs, their probability of heteromerization, and the effect of gene LOF on cell proliferation.**

Average CS values of paralogs (heteromer or not heteromer) belonging to a gene set were used in the analysis. On the *y*-axis, GO terms for molecular functions are sorted according to their proportion of heteromeric paralogs (i.e., # of heteromers/# of paralogs, heteromers defined by "all PPI"). The size of the circles represents the number of paralog pairs in a category, and the colors represent the proportion of heteromers in that category. In the left panel, average CS values of heteromers per category are shown on the *x*-axis. In the right panel, the difference between the average CS value of the heteromers and average CS values of the non-heteromers are shown on the *x*-axis. The terms with significant difference between the average CS value of the heteromers and average CS value of the non-heteromers (estimated by two-sided *t*-test) are annotated with the blue edges. The descriptions of the representative significant GO terms with the highest difference are shown in the right-side panel. Spearman rank correlation between the proportion of the heteromers in the GO terms and the average CS value of paralogs in the term [$r_s$(# of heteromers/# of paralogs per term, CS mean of paralogs per term)] is shown in the lower left corner. Only GO molecular functions with more than 10% of the number of paralogs in all the gene sets are shown. Similar analysis for the GO biological process and GO cellular component aspect, for the "all PPI" based data, is shown in Fig EV2. Similar analysis with the "direct PPI" data is shown in Appendix Fig S6. See Dataset EV5 for GO terms and annotations shown on this figure. Note that not all gene sets are independent because some genes are in several categories.

Source data are available online for this figure.

This shows that some functions are particularly deleterious when deleted and these tend to be rich in heteromeric paralogs. This is the case in the analysis while considering both PPI methods (see Fig 3 for "all PPI" and Appendix Fig S6A for "direct PPI"). The negative correlations also hold true in the case of GO biological process and GO cell component gene sets (see Fig EV2 for "all PPI" and Appendix Fig S6B and C for "direct PPI").

Some molecular functions enriched among heteromers also show a significant difference in the average CS values of the heteromers and non-heteromers in that particular gene set (Fig 3, Dataset EV5). Such gene sets include, for instance, RNA polymerase II and transcription, and DNA and nucleic acids binding genes, which are frequent among large families of dimeric transcription factors that evolve through duplication (Amoutzias *et al*, 2008), some of which have been shown to have co-dependent evolution (Baker *et al*, 2013). Among other such gene sets, phosphatase activity related genes were identified in the case of "all PPI" dataset (Fig 3). Gene sets corresponding to protein homodimerization activity were commonly found in both the analyses with all PPI and with direct PPI (Fig 3 and Appendix Fig S6A) as showing significant lower CS values for heteromeric pairs, consistent with the correlation observed above between heteromerization and homomerization.

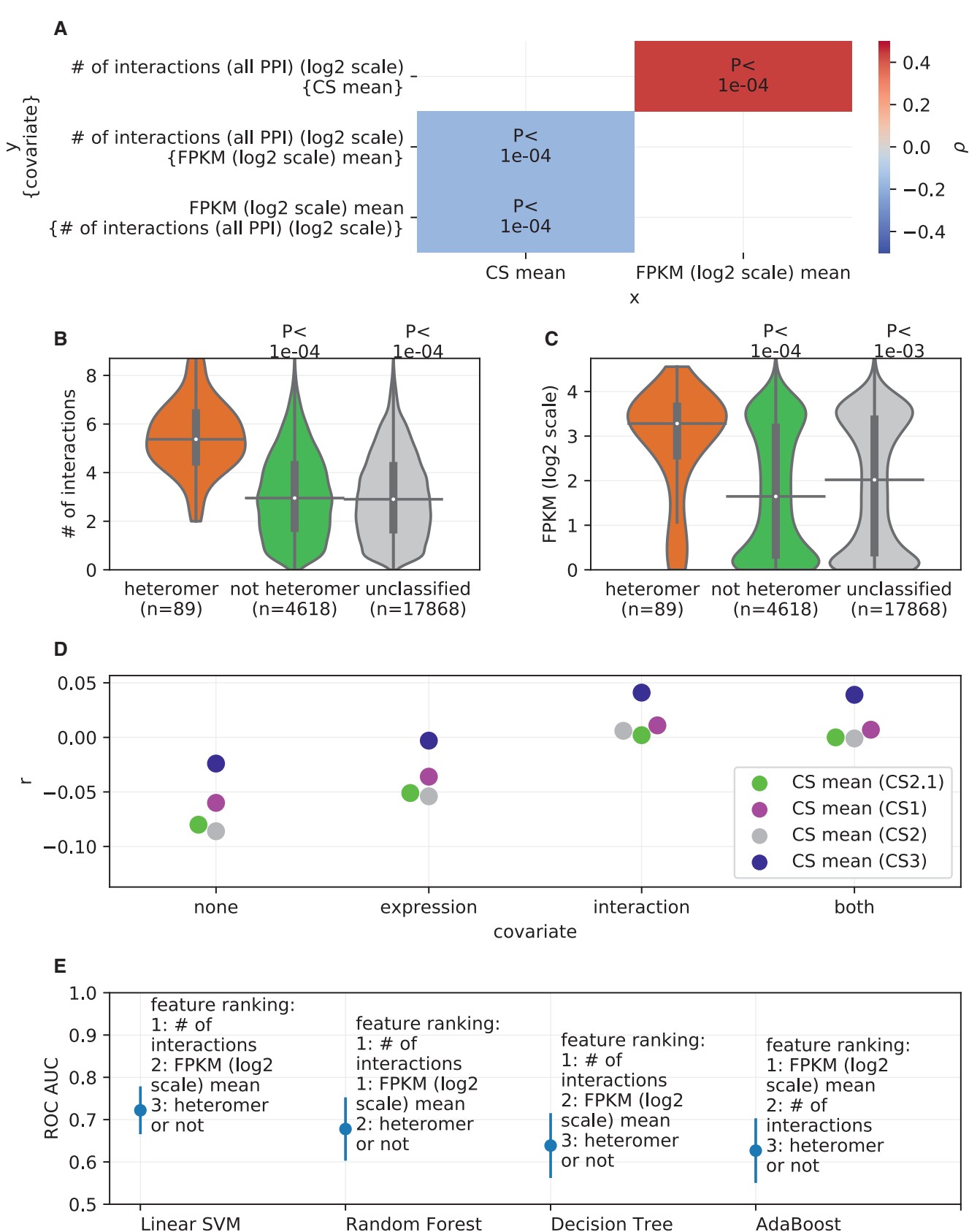

Figure 4.

**Figure 4. Relationship between the effect of LOF of a gene on cell proliferation, mRNA expression, and number of protein–protein interaction partners.**

A  The effect of gene LOF on cell proliferation as measured in terms of CS values is correlated with mRNA expression and number of PPI partners. Considering the interdependence between the three related factors, partial correlations were estimated using Spearman correlation coefficients ($\rho$) between each pair of factors while controlling for the third factor (covariate, indicated in the curly brackets). The P-values associated with the correlations are denoted on the heatmap. Average CS values across CS datasets were used. See Appendix Fig S7 for correlations in case of individual CS datasets and direct PPI.

B  Paralogs that form heteromers have more interacting partners compared to non-heteromers. Number of interactions is in $\log_2$ scale. Similar plot with heteromeric paralogs detected with only direct PPI is shown in Appendix Fig S8A.

C  Paralogs that form heteromers show higher expression than non-heteromers. Similar plot with heteromers of paralogs detected with only direct PPI is shown in Appendix Fig S8B. Cell-line-wise comparisons with heteromers defined by "all PPI" and "direct PPI" are shown in Appendix Fig S8C and D, respectively. Contribution of the interacting factors in determining the paralog status is determined by jointly modeling through two approaches: partial correlations (panel D) and classification models (panel E).

D  Partial Spearman correlation coefficients (r, shown on the y-axis), between CS values and a paralog status (heteromer or not, binary variable, 1: heteromer, 0: not heteromer). The correlations were determined while controlling for none of mRNA expression and number of interactions ("none"), only mRNA expression ("expression"), only number of interactions ("interaction"), or both ("both") (as shown on the x-axis). Controlling for the number of interactions leads to the greater loss of negative correlation, indicating that it contributes to the correlation more than mRNA expression. Similar analysis with heteromers defined by "direct PPI" is shown in Appendix Fig S8E.

E  Feature importance (shown on the y-axis) of the three factors as determined through four different classification models (shown on the x-axis). Means and standard deviations of the ROC AUC values across all cross validations and bootstrapping runs (see Materials and Methods) are plotted for each of the four classifiers. The CS values used for this analysis are mean of the CS values across all the CS datasets. For similar analysis with the four individual CS datasets, see Appendix Fig S9 A–D.

Data information: In panels (B and C), P-values from two-sided Mann–Whitney U tests are shown. On the violin plots, the medians of the distributions are shown by a horizontal black line and quartiles by a vertical thick black line. For clarity, the upper and lower tails of the distributions are not shown.
Source data are available online for this figure.

In terms of biological processes, protein dephosphorylation process-related genes, regulation of cell proliferation, apoptotic, transcription process, and cell junction assembly process are enriched among heteromers and also show significant deleteriousness (i.e., depletion in CS values of the heteromers, see Fig EV2A for analysis with "all PPI", Appendix Fig S6B for analysis with "direct PPI"). In terms of cellular components, essential genes related to actin cytoskeleton and chromatin are enriched among heteromers and are significantly more deleterious (see Fig EV2B for analysis with "all PPI" and Appendix Fig S6C for analysis with "direct PPI"). These genes are therefore interesting candidates for future functional analysis on the consequences of heteromerization for protein function and robustness.

## Heteromeric paralogs are more highly expressed and have more protein interaction partners

From correlations between CS values (from the three biologically independent datasets), mRNA expression, and number of PPI partners, CS values were found to be negatively correlated with the number of PPI partners of a protein and its mRNA expression level (measured in terms of Fragments Per Kilobase of transcript per Million mapped reads, i.e., FPKM; Fig 4A, see Appendix Fig S7 for analysis with each CS dataset). Therefore, it is possible that the deleteriousness of the heteromeric paralogs is partially explained by general dependence of CS values on mRNA expression and number of PPI partners.

Previous reports have shown that homomeric proteins tend to have a larger number of interaction partners (Ispolatov, 2005). If heteromers of paralogs inherit their interactions from the homomeric ancestor, they could also have a larger number of interaction partners, which could explain their relatively lower CS values (Fig 2A). Comparing the number of PPI partners of heteromeric paralogs and non-heteromeric ones, it is clear that heteromeric paralogs have a larger number of PPI partners, both considering "all PPI" (Fig 4B) and "direct PPI" (Appendix Fig S8A) and are more highly expressed (see Fig 4C for analysis with the heteromers

defined by "all PPI" and Appendix Fig S8B for the ones defined by "direct PPI"s). The trend with mRNA expression is also true in most of the cell lines (see Appendix Fig S8C for analysis with heteromers defined by "all PPI" and Appendix Fig S8D for analysis with "direct PPI" only). Collectively, the number of PPI and mRNA expression seem to explain the greater deleteriousness of the heteromers compared to non-heteromeric paralogs.

Further, we tested the extent to which heteromeric status of the paralog can predict the deleteriousness relative to other potential predictors, i.e. mRNA expression and number of interaction partners. In order to do this, considering that the molecular features are interdependent, we relied on two joint modeling approaches based on (i) partial correlations and (ii) machine learning to estimate the predictiveness of the molecular features, as detailed below.

Firstly, the partial correlations were carried out between the deleteriousness of the paralog (CS values) and the status of the paralog being either heteromer or not (binary variable), while controlling for either mRNA expression or the number of protein interaction partners (Fig 4D, for analysis with "direct PPI"s see Appendix Fig S8E). From this analysis, it is apparent that mRNA expression and number of PPI partners are better predictors of deleteriousness relative to heteromeric state of the paralogs, as controlling for each of the two molecular features diminishes the correlation coefficient. Also, between mRNA expression and number of PPI partners, the number of protein interaction partners is a better predictor of the deleteriousness of paralogs than mRNA expression, because controlling for the former diminishes the correlation more than controlling for the later.

In the second joint modeling approach, we used a set of four machine learning classification models to predict the deleteriousness of the paralog, using the three features: (i) heteromeric state of the paralog (heteromer or not, binary variable), (ii) mRNA expression, and (iii) the number of protein interaction partners (see Materials and Methods). From the feature importance obtained from the classification models, it is again apparent that the number of interactions of a protein is a likely better predictor of the status of the paralog (Fig 4E, for analysis with individual CS datasets, see Appendix Fig S9A–D), and thereby of their relative deleteriousness.

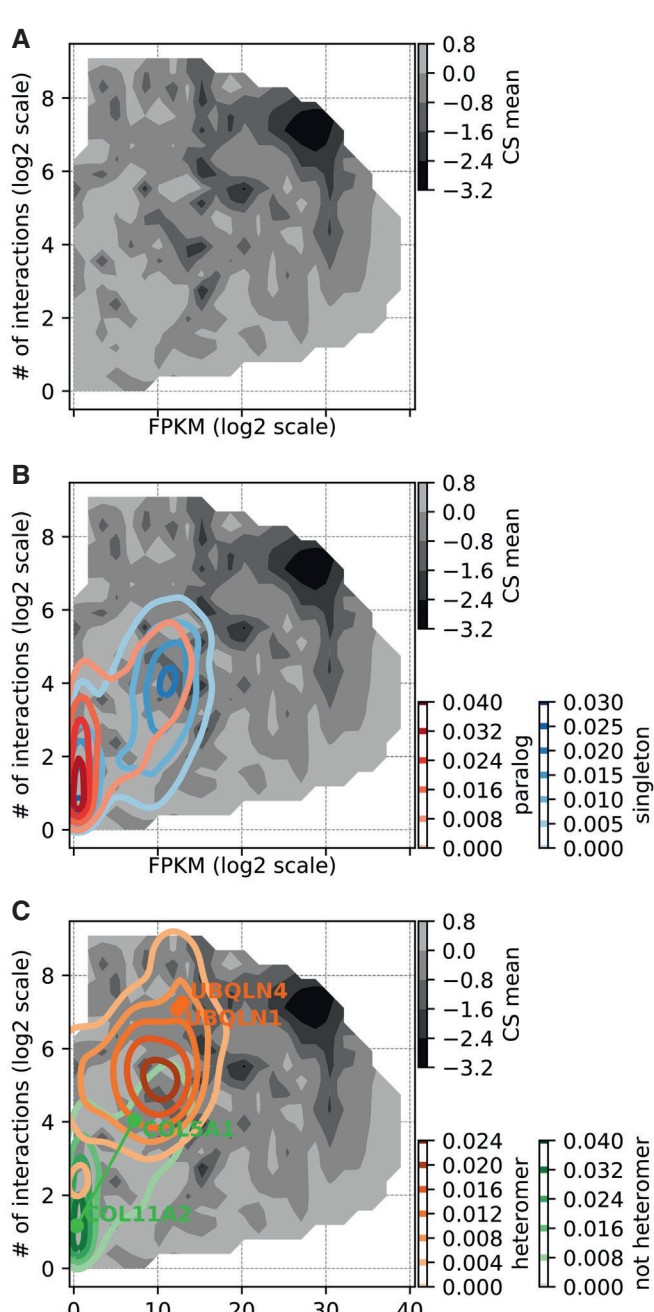

**Figure 5. Robustness landscape visualization showing regions of deleteriousness to LOF as a function of mRNA expression and number of interaction partners.**

mRNA expression (lined on x-axis) and number of PPI partners (y-axis) are strong determinants of the deleteriousness of gene LOF (measured in terms of average CS across CS datasets, shown on z-axis).

A  The landscape shows the effect of LOF of genes on cell proliferation (CS) as a function of the two parameters. The region with high gene expression levels and large number of interactions clearly shows relatively lower CS values, indicating greater deleteriousness upon LOF.

B  Kernel density estimates for paralogs and singletons are overlaid on the landscape to indicate their level of occupancy. The density of paralogs is located toward lower expression levels and small numbers of protein interaction partners, compared to singletons.

C  Similar to (B), kernel densities of heteromeric paralogs and non-heteromeric ones are overlaid on the landscape. The location of heteromers is biased toward higher expression levels and larger number of protein interaction partners, compared to non-heteromers. Also, locations of representative heteromeric (UBQLN1 and UBQLN4) and non-heteromeric pairs (COL5A1 and COL11A2) are annotated on the landscape.

Data information: Similar plots with direct PPIs only are shown in Fig EV4. Source data are available online for this figure.

expression of the genes, but this does not completely explain the results. In the case of all the CS datasets, at lower expression values, the difference in CS between paralogs and singletons is non-significant (Fig EV3E). The reason for this could be attributed to the small counts of mRNA expression generally being relatively more noisy as well as lower fitness effects in general, making differences more difficult to detect. Also, sequence similarity between the paralogs leads to removing reads from the analysis and may thus act as one of the confounding factors in this analysis (see Materials and Methods) by underestimating mRNA expression of paralogs.

### Heteromerizing paralogs occupy a space of the robustness landscape where gene LOF is more deleterious

Overall, these results can be summarized in a robustness landscape in which the robustness against LOF is shown as a function of the number of PPIs and mRNA expression level. Following the pattern of correlations between the three factors (as shown in Fig 4A), the landscape clearly shows that strong deleteriousness is localized in the upper right corner, where expression and number of interaction partners are high (Fig 5A for analysis with "all PPI", see Fig EV4A for analysis with "direct PPI" only). The overlay on this landscape of the density of singletons and paralogs shows that they occupy a similar space (Fig 5B analysis with "all PPI", see Fig EV4B for analysis with "direct PPI" only), although singletons are in a space in which genes are slightly more expressed and have a slightly larger number of interactions. Paralogous genes that form heteromers clearly occupy a space that is distinct from non-heteromeric ones, which brings them closer to a more deleterious parameter space, in which genes are relatively more expressed and have more protein interaction partners (Fig 5C for analysis with "all PPI", see Fig EV4C for analysis with "direct PPI" only). We also show representative pairs of heteromeric and non-heteromeric pairs of paralogs on the map (Figs 5C and EV4C). For instance, Ubiquilin 1 and 4 (UBQLN1 and UBQLN4) form a heteromer, are highly expressed, and have a large number of protein interaction partners and their LOF is highly deleterious, as seen by their location nearing the valley of the fitness landscape. On

In addition, multiple regression analysis (Appendix Fig S9E and F) also corroborated these results. Put together, these analyses thus confirm that indirect effects owing mostly to the number of PPI partners, and to mRNA expression to a lesser extent, seem to explain the stronger impact of LOF on heteromers.

As an addendum, in general, the protective effect of paralogs compared to singleton is partially caused by their lower expression and smaller number of protein interaction partners than that of the singletons (Fig EV3A and B, cell-line-wise comparison in Fig EV3C). Partial correlation analysis (Fig EV3D) indicates that the protective effect of the paralogs is more attributable to the

the other hand, a non-heteromeric pair, collagen type V alpha 1 chain (COL5A1) and collagen type XI alpha 2 chain (COL11A2) have lower mRNA expression and lower number of PPI partners, setting their position at the peak of the landscape with relatively non-deleterious CS values. In terms of network features, paralogous pairs that heterodimerize are more similar to singletons and have correspondingly similar effects on proliferation when inactivated.

## Potential consequences of the heterodimerization of paralogs

We examined the results from the meta-analysis further to explore other potential features leading to the association between the

heteromerization of paralogs and their deleteriousness upon LOF. Given that gene expression level appears to be one of the determinants of the fitness effect of LOF, it is possible that mechanistically, the ability of a paralog to buffer for the loss of its sister copy depends on their relative abundance. For instance, if highly asymmetrically expressed, the LOF of the most expressed gene of a pair is unlikely to be buffered by the least expressed one. However, if both are expressed at a comparable level, both would be expected to affect cell proliferation in a comparable manner. In addition, Diss *et al* (2017) showed that physical dependency between interacting paralogs is often asymmetrical, the lowly expressed copy being affected by the deletion of the highly expressed one more than in

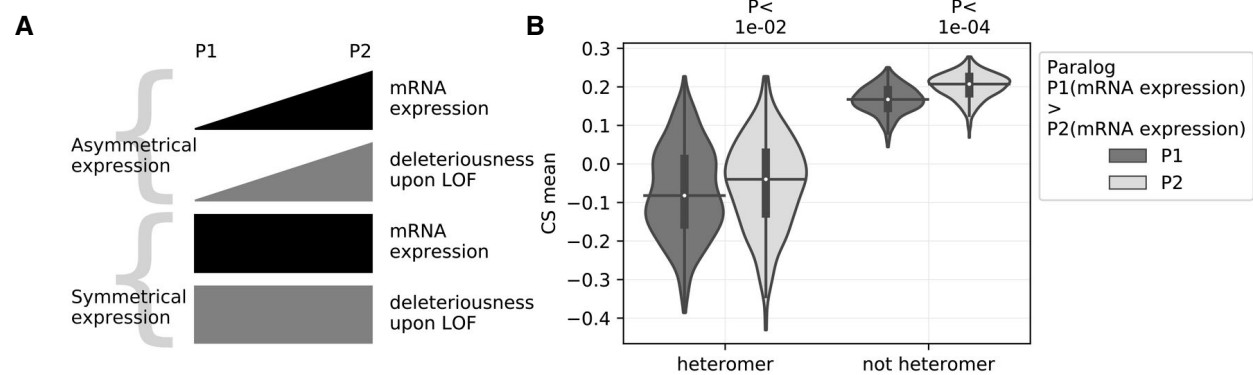

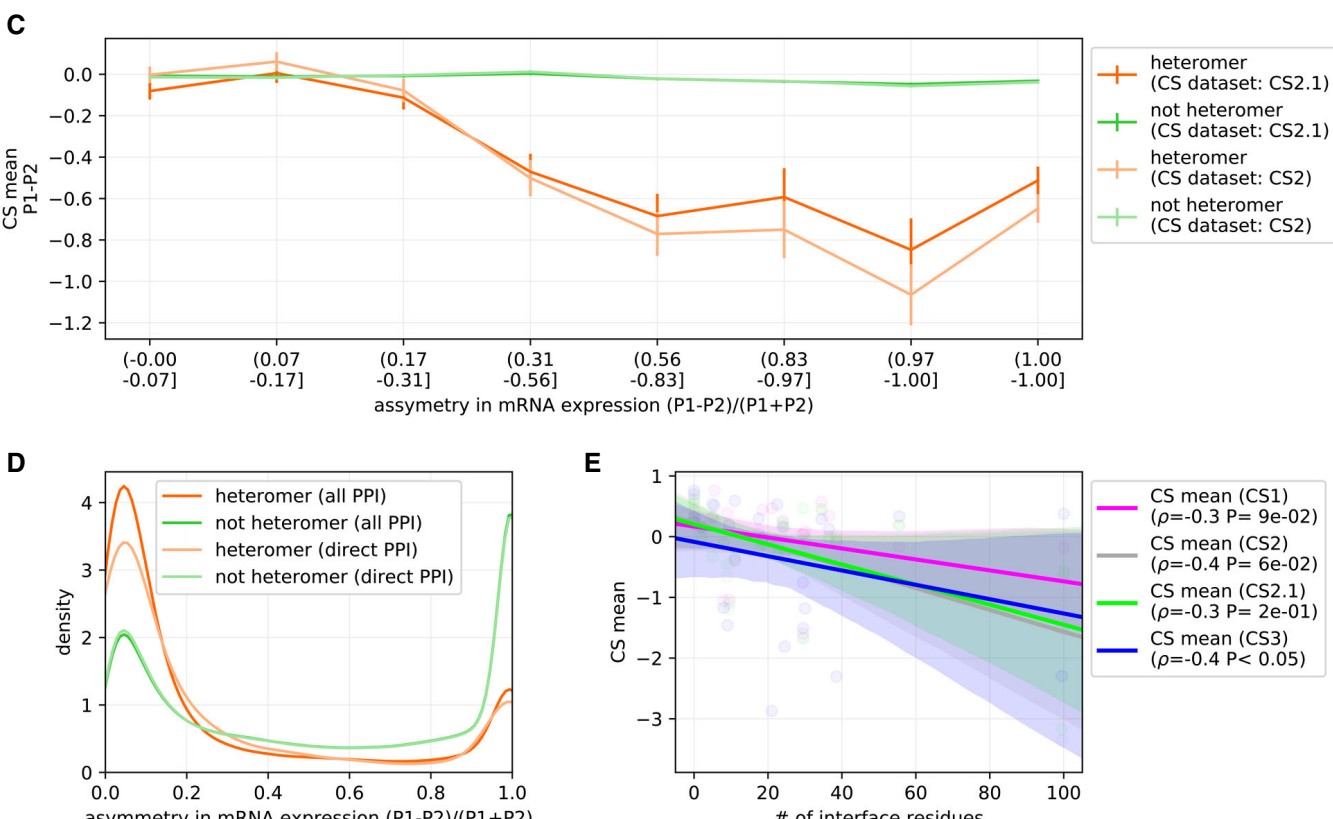

**Figure 6.**

**Figure 6. Asymmetric expression of paralogs and mechanistic insights into the relatively greater deleteriousness of heteromeric paralogs.**

A  Schematic representing likely scenarios pertaining to the relationship between the asymmetry in mRNA expression of a pair of paralogs (P1 and P2) and their relative deleteriousness upon LOF, as discussed in the text.

B  The most expressed paralog (P1) of a pair is more likely to be deleterious than the least expressed (P2). mRNA expression data is composed of 374 cell lines. Each point represents CS value of an individual cell line. *P*-value is from two-sided Mann–Whitney *U* test. On the violin plots, the medians of the distributions are denoted by a horizontal black line, and quartiles of the distributions from the medians are indicated by a vertical thick black line. For clarity, the upper and lower tails of the distributions are not shown. Heteromers in this analysis are defined from the "all PPI"s. For similar analysis with "direct PPI"s, see Appendix Fig S11.

C  Relationship between the difference in CS of the paralog pair (P1 − P2) and the asymmetry of mRNA expression levels, i.e., (P1 − P2)/(P1 + P2), where mRNA expression of P1 is higher than P2. The values of asymmetry of mRNA expression levels close to 0 are cases in which the mRNA expression is symmetrical and asymmetrical for values near 1. Error bars represent 95% confidence interval with respect to difference in CS (*y*-axis) of the paralog pairs defined by equal sized bins of the asymmetry of mRNA expression (*x*-axis). The heteromers are defined by "all PPI". Similar analysis with heteromers defined by "direct PPI" is shown in Appendix Fig S13A. The relationship between the two factors in case of representative pairs of heteromeric and non-heteromeric paralogs is shown in Appendix Fig S14. Comparison of distributions of the correlation scores between heteromers and non-heteromers is shown in Fig EV5B.

D  Heteromeric paralogs tend to have more symmetric mRNA expression as compared to non-heteromers. Distribution of the asymmetry in the mRNA expression, i.e., (P1 − P2)/(P1 + P2), where mRNA expression of P1 is higher than P2. The values near 0 are cases in which the mRNA expression is symmetrical and asymmetrical for values near 1.

E  The deleteriousness of the heteromers upon LOF (lined on the *y*-axis) is negatively correlated with the number of residues at the interaction interface (*x*-axis). $\rho$ is Spearman's correlation coefficient. *P*-values associated with the Spearman's correlation coefficient are shown in the legend. Structures of representative heteromers are shown in Appendix Fig S15.

Source data are available online for this figure.

the reciprocal condition. They suggested that this asymmetry could derive from the fact that the lowly expressed one may be post-translationally stabilized by the most expressed one. We therefore tested whether asymmetry in mRNA expression influences which gene in a pair is more deleterious upon LOF (Fig 6A).

We found that across cell lines, the paralog of a pair with the highest level of expression (P1) shows significantly lower CS values than the lowly expressed one (P2) (Fig 6B for analysis with heteromers defined by "all PPI", see Appendix Fig S11 for similar analysis with heteromers defined by "direct PPI"). This means that the loss of a paralog is more deleterious when it is the most expressed one in a pair. Properties that determine the deleteriousness of LOF across genes therefore also apply within pairs of paralogs. The fraction of pairs across cell lines for which the LOF effect is greater than the other paralog in the pair is strongly associated with their asymmetry of expression (Fig EV5A for analysis with CS2.1 dataset, see Appendix Fig S12 for similar analysis with CS2 dataset). Moreover, investigating the relationship between the difference of average CS of the paralog pair and asymmetry of mRNA expression levels, we find that these two factors are more negatively correlated in the case of heteromers than non-heteromers (see Fig 6C for analysis with "all PPi" and Appendix Fig S13A for analysis with "direct PPI". See Fig EV5B and Appendix Fig S13B for distributions of correlation coefficients in case of CS2.1 and CS2 datasets, respectively), although the difference is statistically significant only in the case of heteromers defined by "all PPI" and CS2.1 dataset. These correlations indicate that if the heteromeric pairs of paralogs are asymmetrically expressed, then the difference in deleteriousness is larger upon LOF than for non-heteromeric ones. This suggests that the lowly expressed gene of a pair is less able to buffer the loss of the highly expressed one in case of heteromers than non-heteromers. This enhanced sensitivity appears to be counterbalanced at the systems level by the fact that heteromeric paralogs are more likely to have symmetrical expression (Fig 6D). Altogether, this suggests that symmetrical expression imposes a stricter contingency for heteromers than non-heteromers, arguably due to the need for their stoichiometric balance in their physical assembly. Comparing the mRNA expression of the heteromers across 374 cell lines

(Fig EV5C) and protein expression across 49 cell lines (Fig EV5D), it appears that the heteromers are indeed on average more dosage balanced than non-heteromers. This trend is observed in both the comparisons and data from either PPI source, although it is statistically significant only in the case of comparison of mRNA expression for heteromers defined by "all PPI". This is potentially because it covers more pairs or because those include more pairs in large complexes, which are submitted to dosage balance constraints (Papp *et al*, 2003; Teichmann & Veitia, 2004), and because the proteomics data cover a smaller number of gene pairs and cell lines.

Finally, we examined whether the enhanced deleteriousness of LOF for heteromers could be due to their physical dependency, which would be manifested as the alteration of one member of a pair when the other member is absent as previously observed by Diss *et al* (Pickett & Meeks-Wagner, 1995; Diss *et al*, 2014). This could offer a mechanistic insight into some of our observations. It is difficult to predict the physical dependency of paralogs, but one could hypothesize that it is more likely to occur for strongly interacting pairs. We therefore used the size of the interaction interface of heteromers as a proxy for the strength of interaction [as in (Sousa *et al*, 2019), see Materials and Methods]. Using the data for 25 heteromers of paralogs, we indeed observed a marginally significant negative correlation with the average CS values (Fig 6E) of paralog pairs with the strength of interactions, suggesting that codependency indeed could be a mechanism that contributes to the enhanced deleteriousness of paralogs pairs that interact with each other.

## Discussion

The contribution of gene duplicates to cellular robustness has been established for several individual genes prior to the era of large-scale screening (Thomas, 1993; Melton, 1994; Pickett & Meeks-Wagner, 1995; Gibson & Spring, 1998). It was well established for model organisms such as yeast for which systematic gene deletion experiments have been performed (Gu *et al*, 2003). Systematically

investigating the extent of the contribution of gene duplication to cellular robustness in the context of human cells was only recently made possible owing to large-scale CRISPR-cas9 screening (Wang *et al*, 2015). Here, using three biologically independent datasets of gene LOF that represent a large number of diverse cell lines and different experimental approaches (Table EV1), we find that paralogs systematically contribute to cellular robustness across all cell lines (Fig 1A).

Although the signal for the contribution of gene duplicates to robustness is significant and reproducible across datasets, some factors could limit the effects measured. The types of the cell lines used, i.e., cancer cell lines (in case of CS1, CS2/2.1) and primary (in case of CS3), clearly show a difference in terms of the correlations (Appendix Fig S1). A second factor that is particular to the LOF screens in mammalian cell lines is the robustness of these cells to gene LOF. As seen from the distributions of the CS values across CS datasets, most of the genes are robust to LOF (Fig EV1). The effective range of deleteriousness is thereby very narrow, limiting the resolution of the comparisons. Another limiting factors is that not all of the paralogs could be present as pairs in all cell lines, in particular in cancer cell lines that may have had additional duplications and deletions, which may have altered the copy number of paralogous genes. Although this effect may have biased our analyses, the use of the dataset CS2, which has been corrected for copy-number variation across the cell-line genomes (Meyers *et al*, 2017), shows that the results are likely robust to these effects. Another factor that may affect the results is that gRNAs could potentially inactivate both paralogs of a pair, thereby leading to double gene LOF rather than a single one (Fortin *et al*, 2019). For instance, the CS1 original dataset (Wang *et al*, 2015) contained gRNAs that targeted multiple positions in the genomes, many of which could be positions that correspond to duplicated genes. This could lead to double gene LOF by Cas9 cutting and DNA repair but could also lead to chromosomal rearrangements, leading to even stronger effects (Després *et al*, 2018; Kosicki *et al*, 2018) than double gene LOF (Fortin *et al*, 2019). For this reason, we re-analyzed all data and considered only uniquely aligned gene-specific gRNAs (see Materials and Methods). Nevertheless, it is not clear how many mismatches could be tolerated for efficient mutagenesis by Cas9 activity to occur, so eliminating all gRNAs that could lead to more than one gene LOF remains a difficult task.

We examined whether specific features of paralogous genes could affect their ability to buffer each other's LOF effect. We focused on their heteromerization because recent reports have shown that paralogous proteins often physically associate and that these physical associations could reduce their ability to buffer each other's LOF (DeLuna *et al*, 2010; Diss *et al*, 2017). This observation led to the prediction that due to their physical and thus potential functional dependency, paralogs that form heteromers could contribute less to cellular robustness than non-heteromers, essentially behaving like singletons. We found that these paralogs indeed lead on average to larger effects on cell proliferation when inactivated by LOF mutations (Fig 2A). However, this is largely explained by larger number of protein interaction partners and higher expression levels for this class of paralogs (Fig 4). On the robustness landscape outlined by the two factors (Fig 5A), expression levels and numbers of protein interaction partners clearly separate genes based on their deleteriousness. It also helps in understanding the major

determinants of buffering effect of paralogs in general (Fig 5B) and the greater deleteriousness of heteromers (Fig 5C).

One limitation of our analysis is the use of physical interactions between paralogs as a proxy for dependency. Indeed, physical interactions may not be necessary nor sufficient for paralogs to be dependent (Kaltenegger & Ober, 2015). It is possible that dependency concerns only obligate heterocomplexes, which are difficult to distinguish in large-scale data. However, our analyses using protein interaction interface size as a proxy for interaction strength suggest that this could be a potential mechanism. Dependency between paralogs could also evolve by other means than physical interactions. Additionally, it is also difficult to determine from large-scale proteomics data if two paralogs are part of the same complex simultaneously or if they occupy the same position but switch according to cellular compartments of expression timing (Ori *et al*, 2016). In this latter case, it would be unlikely that paralogs are dependent on each other, although the proteomics data would inaccurately suggest that they physically interact by being in the same complex. Finally, there is also an issue with the sparseness of the known interactome. For instance, we still lack evidence for direct physical interactions (i.e., direct PPI) for most cases. This eventually obscures analysis because of lack of statistical power (as in the case of Appendix Fig S5).

Our results show that the association between paralog heteromerization and strong fitness effects is largely if not completely driven by the fact that it is also associated with expression levels and number of protein interaction partners. This is in line with the observation made by Wang *et al* (2015) who showed that essential genes tend to be more expressed and have a larger number of protein interaction partners. Recent observations supporting this trend were made by showing that LOF variants are rarer in humans for proteins with large number of protein interaction partners (preprint: Karczewski *et al*, 2019). Here, we observe a similar result and identified that such features are enriched among paralogs that form heteromers. It is therefore difficult to determine if heteromerization indeed prevents buffering directly because of cross-dependency, or if all of the effects measured are caused by abundance itself. Our analysis showed that heteromeric paralogs have a tendency to be often associated with particular molecular functions (Fig 3) and these functions appear to lead to stronger effects on cell proliferation when inactivated. Heteromers of paralogs could therefore also have a lower buffering capacity overall because they associate with specific functions, including, for instance, transcription factors and protein kinases.

The capacity of paralogs to buffer each other's LOF also likely depends on their mechanisms of maintenance, which can be for instance subfunctionalization or neofunctionalization (Force *et al*, 1999; Lynch & Force, 2000; Lynch *et al*, 2001; Innan & Kondrashov, 2010). Which one applies here for each paralog pair is difficult to determine without knowledge of the ancestral functions of the genes prior to duplication. Paralogs could fall into three categories. First, the duplication could be mostly neutral and has not been maintained by natural selection. Under this scenario, and in the absence of other changes, gene duplicates should be able to compensate each other's loss as long as they persist. Their function should not depend on each other's. The second possibility is that one copy or the other or both have neofunctionalized (reviewed in Innan & Kondrashov, 2010). In this case, the novel function acquired by a paralog could not be compensated by the other copy but all

ancestral functions could. Dependency in this case could arise from the acquisition of new functions by both paralogs at the same time. This has been seen, for instance, by Boncoeur *et al* (2012) who showed that the drug-pumping specificity of some ABC transporters are specific to heterodimers of paralogs and cannot be performed by individual homodimers. Buffering of this function would be possible by neither of the paralogs. For transcription factors, the neofunctionalization of one copy could be to become a repressor of the second copy, essentially given the heteromer a new functionality (Bridgham *et al*, 2008). The heteromer would now have a new regulatory mode that depends on the presence of both paralogs.

The final scenario is often supported for the maintenance of paralogs and involves the accumulation of complementary degenerate mutations that lead to subfunctionalization (Force *et al*, 1999; Lynch *et al*, 2001). In this case, paralogs are maintained but without a net gain of function. This degeneracy would prevent compensation for the functions that have been lost in one or the other paralog. However, any function that did not subfunctionalize could be compensated for the second paralog upon the deletion of the first one. One way subfunctionalization could lead to dependency would be by complementary degenerate mutations (Kaltenegger & Ober, 2015) that maintain the heterodimer but lead to the loss of the homomers when proteins need to act as multimers (Pereira-Leal *et al*, 2007). In this case, the heteromeric form could replace the homomers while making the presence of both paralogs necessary and preventing their mutual compensation. Dependency is therefore compatible with several modes of paralog maintenance but how frequent it is in each case remains to be examined and may require detailed functional characterization of paralog pairs.

We found that the relative expression level of paralogs is significantly associated with which one would be the most deleterious upon LOF (Fig 6A and B), revealing that buffering capacity is dependent on relative expression levels. Consistent with our observation, Barshir *et al* (2018) recently showed that genetic diseases that are tissue specific and that affect paralogous genes tend to affect tissues in which the second copy of a pair is generally lowly expressed, reducing its buffering capacity. These observations and ours have important consequences regarding the buffering effects of paralogs and their evolution. Qian and Zhang (2008) and Gout *et al* (Gout *et al*, 2010; Gout & Lynch, 2015) showed that expression levels alone could be a strong determinant for the maintenance of paralogous genes. According to their model, paralogs would drift from one another in terms of expression levels (Gu *et al*, 2002) because only their cumulative abundance is gauged by natural selection. Functional divergence at the protein level would therefore not be necessary for paralogs in order to lose their buffering ability, and divergence of expression would be enough. Once a paralog is dominating expression level, the loss of the second copy becomes almost inconsequential, rendering its loss effectively neutral. Our results support this model by showing that the loss of the least expressed paralog of a pair is generally less consequential than the loss of the most expressed ones (Fig 6B). If gene expression levels evolve at a faster rate than protein functions, this type of subfunctionalization could be the dominating cause of paralog maintenance and at the same time contribute largely to shape the robustness landscape of cells to LOF mutation. Interestingly, we found that in general, heteromerizing pairs of paralogs have more symmetrical expression levels than non-heteromerizing ones (Fig 6D). Heteromerization

could slow down gene expression drifting and contribute to paralog maintenance, which could explain the relatively older age (higher dS) of heteromers (Fig 2D). Interestingly, for heteromerizing ones, the difference of deleteriousness between paralogs is strongly correlated with their asymmetry in mRNA expression (Figs 6C and EV5B), indicating that there are larger differences in terms of the deleteriousness when the paralogs have different abundances. In addition, maintenance of better dosage balance through regulation at the transcriptional and post-transcriptional levels (Fig EV5C and D) indicates a potential contingency on their stoichiometry, most likely imposed by the requirement for the assembly of the heteromers. Finally, we observed that heterodimers of paralogs could be more dependent on each other if their interaction is stronger. Our results concern a very small set of proteins and uses a proxy for binding strength. They will therefore need to be investigated further.

Overall, our analyses show that not all paralogs are equally likely to buffer each other's LOF in human cells. The underlying mechanisms for this ability remain to be fully understood beyond gene expression and protein–protein interactions and may depend on the specific function of paralogs. Overall, considering the frequent occurrence of copy-number variations in cancer cells, the insights obtained from this study regarding the mechanism of robustness of duplicates could be relevant in the development of cancer therapies. However, more detailed functional analyses will be required to fully determine what is the role of paralog dependency and how dependency could be driven by the physical assembly of paralogs. Since previous studies have shown that this dependency could take place through post-translational regulation, a systematic combination of gene LOF and protein abundance measurements would be the next important step to efficiently identify dependent paralogous genes.

## Materials and Methods

### Protein–protein interactions

The human protein–protein interaction data were obtained from BioGRID (Data ref: BioGRID, 2018; Chatr-Aryamontri *et al*, 2015, 2017) and IntAct (Orchard *et al*, 2014; Data ref: IntAct, 2019). While defining all methods of detection for protein–protein interactions ("all PPI"), co-fractionation, protein-RNA, co-localization, proximity Label-MS, and affinity capture-RNA were removed because they are not strictly speaking capturing PPIs. A subset of these methods capturing "all PPI", defined as two-hybrid, biochemical activity, protein-peptide, PCA and Far Western were considered as methods detecting "direct PPIs". The number of PPI partners per gene is provided in Dataset EV6.

### Gene sets: paralogs and singletons

The set of human paralogs was obtained from the Lan *et al* study (Lan & Pritchard, 2016) and is enriched for small-scale duplication events (1,436 pairs). This set of paralogs was completed with a set of paralogs from whole-genome duplication events, obtained from the Ohnologs-2 database (Singh *et al*, 2015) using the "strictest" set (Data ref: Ohnolog, 2018). As a complete set, 3,132 non-redundant pairs of paralogs were used in the study (Dataset EV1). Only the paralogs for which annotations exist in the Ensembl Compara database (Herrero

*et al*, 2016) were used in the analysis. For the merging of the datasets, gene ids of the paralogs were obtained from both Ensembl release 75 and 95 (Zerbino *et al*, 2018). Protein ids of the paralogs were retrieved from Ensembl Compara (Herrero *et al*, 2016).

Pure singletons were identified using BLASTP (Altschul *et al*, 1990) searches of the unique sequences from human proteome (Data ref: Human proteome sequences, 2018) against itself. Any protein that had no hits with E-value smaller than 0.001 over a segment longer that 0.6 times the smaller protein was considered as singleton. Gene symbols were used to merge the data from paralogs and protein interaction data.

The list of paralogs and singletons is included as Dataset EV1.

### Gene sets: heteromers and homomers

From the protein–protein interactions, heteromers of paralogs were identified as the pairs of paralogs that physically interact with each other. The rest of the paralog pairs were classified as "not heteromer". Homomers are proteins that interact with themselves. This classification was carried out considering both "all PPI" and "direct PPI". The number of homomers and heteromers identified by each method is indicated in Appendix Table S1.

The gene sets (i.e., heteromers and homomers) identified through PPI from BioGRID and IntAct datasets were merged by taking intersections. For instance, heteromers identified in both datasets were considered in the merged dataset. If the dataset (BioGRID or IntAct) is not mentioned, the merged dataset is used in the given analysis.

The list of heteromers and homomers is included as Dataset EV1.

### Gene sets: essential and non-essential genes

Sets of essential and non-essential genes were derived from the union of gene sets reported by DepMap (2018) and BAGEL (Hart & Moffat, 2016).

The list of essential and non-essential genes is included as Dataset EV1.

### Sequence divergence scores and age groups of paralogs

dS scores were estimated using codeml (Yang, 2007). Protein sequences and coding sequences (CDS) of the paralogs were obtained from GRCh38 assembly of the human genome (Ensembl genome version 95), using pyEnsembl (Rubinsteyn *et al*, 2017). dS value greater than 5 was not considered in the analysis (Fig 2D), because larger values are likely saturated and non-reliable.

The age groups of the paralogs, i.e., evolutionary distances in terms of the taxonomy levels, were retrieved from Ensembl Compara (Herrero *et al*, 2016). The evolutionary distances of taxonomy levels were obtained from the Ensembl species tree (Ensembl Species Tree https://ensembl.org/info/about/species.html).

dS values, age groups, and evolutionary distances of the age groups are included in Dataset EV1.

### CRISPR score dataset CS1 (Wang *et al*, 2015)

The CS values of set CS1 were derived from data from genome-wide CRISPR-Cas9 screening experiments from Wang *et al* (2015). The

raw sequencing read counts were re-analyzed to remove all gRNA that hit more than one locus in the genome (multi-hit gRNAs) as these could possibly lead to double gene knockouts, particularly for young paralogs. For the cell lines with replicated experiments, replicates of the read count data were averaged. The resulting raw read counts were used as input of BAGEL (Hart & Moffat, 2016) to calculate the fold changes. The fold changes calculated by BAGEL were then multiplied by −1 (in order to scale them according to the gene essentiality), so that lower values indicate higher relative deleteriousness. Z-score normalized fold-change values were used as CS values per gene. Gene-wise CS values from the CS1 dataset are included in Dataset EV3.

### CRISPR score dataset CS2 (DepMap, 2018)

We used the published data from the DepMap consortium (DepMap, 2018) (18Q3 release) that corresponds to genome-wide CRISPR knock-out screen in cancer cell lines. The CS values in this case are corrected for copy-number variation using CERES (Meyers *et al*, 2017). A total of 450 cell lines with replicated experiments were considered in this dataset (Table EV1). CS values obtained from the DepMap repository (DepMap, 2018; file name: gene_dependency.csv) were z-score normalized and integrated in the overall CS dataset. Gene-wise CS values are included in Dataset EV3.

### CRISPR score dataset CS2.1 (DepMap, 2018)

The CS2.1 dataset was generated by analyzing data for the same experimental system as CS2 (DepMap, 2018) (18Q3 release) but with the removal of "multi-hit" gRNAs that may lead to double paralog knockouts. gRNA-wise fold-change values (file name: logfold_change.csv) were used in the analysis. The associated gRNA to gene map (file name: guide_gene_map.csv) was used to obtain gene-wise CS values. The fold-change values per gene were calculated using BAGEL (Hart & Moffat, 2016) as mentioned above. Z-score normalized fold-change values were used as CS values per gene. Gene-wise CS values are included in Dataset EV3.

### CRISPR score dataset CS3 (Shifrut *et al*, 2018)

CS values for the CS3 dataset were obtained from a genome-wide CRISPR-Cas9 screening experiment in primary T cells (Shifrut *et al*, 2018). This dataset serves as an independent reference to the cancer or immortalized cell lines used in the other datasets. gRNAs from the study were first filtered to remove all the multi-hit gRNAs that may lead to double paralog knockouts. CS values per gene were obtained by processing the gRNA counts using BAGEL (Hart & Moffat, 2016) as described above. Z-score normalized fold-change values were used as CS values per gene. The CS3 dataset is included in Dataset EV3.

### Merging of CRISPR score datasets

For the comparative analysis of the four datasets, CS values in each dataset were first quantile normalized and individual datasets were merged by gene symbols (Ensembl release version 75). Cell-line-wise merged CS datasets are available in the BioStudies

database (http://www.ebi.ac.uk/biostudies) under accession number S-BSST233, and aggregated CS values from all datasets are provided in Dataset EV3.

In case of datasets CS2 and CS 2.1, as an aggregated CS value per gene, the average CS value per gene over cell lines was computed. Mean and median aggregations were found to correlate very strongly (pearson's $r \sim 0.99$); therefore, mean aggregation was used. Unless mentioned otherwise, the average CS values across datasets are used as a vector of gene-wise CS values, for instance, in case of Figs 3 and EV2.

### GO enrichment analysis

The GO Molecular Function enrichment analysis was performed using GSEA (Subramanian *et al*, 2005) and Enrichr (Chen *et al*, 2013a; Kuleshov *et al*, 2016) through gseapy tool (https://github.com/zqfang/GSEApy). Note that GO gene sets may originate from evidence which may not entirely be independent from the rest of the data used in the meta-analysis. Therefore, potentially confounding sources of evidence pertaining to the sequence orthology [Inferred from Sequence Orthology (ISO) and Inferred from Physical Interaction (IPI)] were removed from the gene set annotation file.

The list of all the paralogs was used as the reference set, and the list of heteromeric paralogs from the "all PPI" data (analysis shown in Figs 3 and EV2) and from the "direct PPI" (analysis shown in Appendix Fig S6) were used as the test sets. GO term annotations used in the analysis are included as Dataset EV4, and *P*-values are available in Dataset EV5.

### mRNA expression

In order to obtain gene expression levels of paralogs, we used transcriptomics data from the Cancer Cell Line Encyclopedia (CCLE; Barretina *et al*, 2012). We considered data of the 374 cell lines that had complementary CS data in the CS2 and CS2.1 datasets (see Dataset EV2 for cell lines used). Raw RNAseq alignment files (BAM format) were obtained from Genomic Data Commons (GDC) portal (https://portal.gdc.cancer.gov/). Expression divergence of paralogous genes may be underestimated due to their sequence similarity. In order to address this confounding factor, we only considered uniquely aligned reads. Such reads were obtained by filtering the raw BAM files using SAMtools (Li *et al*, 2009) command: "samtools view -bq 254 -F 512 $bamp | samtools rmdup -sS - $bamp.unique.bam". Here, $bamp is the path to the raw BAM file. Next, the FPKM values were estimated using Cufflinks (Trapnell *et al*, 2012): "cufflinks -p 1 –max-frag-multihits 1 -g $gtfp -o output_folder $bamp.unique.bam". Here, $gtfp is path to the annotation file (*Homo sapiens*, assembly: GRCh37, Ensembl release:75) and $bamp.unique.bam is the path to the BAM file containing only unique reads (as made in the preceding step). Gene-wise mRNA abundance is included as Dataset EV7. Cell-line-wise mRNA abundance is available in the BioStudies database (http://www.ebi.ac.uk/biostudies) under accession number S-BSST233.

### Protein expression

Protein expression data for 49 cell lines that are also represented in the mRNA expression dataset and CS datasets were retrieved from Ensembl expression atlas (Papatheodorou *et al*, 2018). This dataset is available in the BioStudies database (http://www.ebi.ac.uk/biostudies) under accession number S-BSST233.

### Classification models

The heteromeric state of the paralogs (either heteromer or not, binary variable), mRNA expression, and number of PPI partners of the protein were used as feature set to predict whether the gene is deleterious or not upon LOF (target). Genes were classified into sets of deleterious and non-deleterious ones on the basis of CS value. The average of the minimum CS value of the non-essential genes and maximum CS value of the essential ones was used as a cutoff to segment the two target classes, i.e., deleterious and non-deleterious genes. Four different classifiers that provide feature importance values were used: Linear SVM, Random Forest, AdaBoost, and Decision Tree. The classifiers were trained using scikit-learn (Pedregosa *et al*, 2011). For training, fivefold cross validations were carried out. In each cross validation, 40% of the data was used as a testing set. For each classifier, default parameters were used to train the models. In order to balance the unbalanced classes, equal-sized data were bootstrapped from the bigger class. ROC-AUC value of a classifier was calculated as an average of all the cross validation and bootstrapped runs.

### Protein interaction interfaces

The size of the interaction interface between heteromeric paralogs was obtained from Interactome INSIDER (Meyer *et al*, 2018). The structures of the interacting paralogs (Appendix Fig S15) were obtained from Interactome3D (Mosca *et al*, 2013).

### Data analysis and visualization

For the retrieval of the CDS and protein sequences, PyEnsembl (Rubinsteyn *et al*, 2017) was used. For mapping of ids, uniprot REST API (UniProt Consortium, 2019) was used. Protein structures were visualized using UCSF Chimera (Pettersen *et al*, 2004). For general statistical analysis, SciPy (Jones *et al*, 2001) was used. Partial correlations were estimated using Pingouin (Vallat, 2018). Plots were generated using matplotlib (Hunter, 2007) and seaborn (Waskom *et al*, 2018) while figures were generated using the Python package rohan (Dandage, 2019). Machine learning modeling was carried out using scikit-learn (Pedregosa *et al*, 2011). The anaconda virtual environment was used to install external programs such as codeml (Yang, 2007).

## Data availability

Cell-line-wise CS values, mRNA expression values, and protein expression were deposited at the BioStudies database (http://www.ebi.ac.uk/biostudies), under accession number S-BSST233. The codes used for the curation of the data and meta-analysis in the study are available at: https://github.com/Landrylab/human_paralogs.

**Expanded View** for this article is available online.

## Acknowledgements

We thank the members of the Landrylab for discussions. We thank Anna Fijarczyk, Philippe Després, Carla Bautista, Johan Hallin, Angel Cisneros, Diana Ascencio, Ugo Dionne, and Axelle Marchant for insightful discussions and comments on the manuscript. RD is funded by Fonds de recherche du Québec-Santé (FRQS) Programme Postdoctoral. This research was supported by the Canadian Institute of Health Research (CIHR) Foundation grant (RN348479—387697) to CRL. CRL holds the Canada Research Chair in Evolutionary Cell and Systems Biology.

## Author contributions

RD & CRL designed and performed research and wrote the paper.

## Conflict of interest

The authors declare that they have no conflict of interest.

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
