## [Review Process File · Molecular Systems Biology]

Paralog dependency indirectly affects the robustness of human cells

Rohan Dandage and Christian R Landry.

Review timeline:	Submission date:	18 th February 2019
	Editorial Decision:	16 th April 2019
	Revision received:	26 th July 2019
	Editorial Decision:	13 th August 2019
	Revision received:	26 th August 2019
	Accepted:	28 th August 2019

Editor: Maria Polychronidou

Transaction Report:

1st Editorial Decision

16th April 2019

Thank you again for submitting your work to Molecular Systems Biology. We have now heard back from the three referees who agreed to evaluate your study. As you will see below, the reviewers acknowledge that the presented conclusions seem potentially interesting. They raise however a series of concerns, which we would ask you to address in a major revision.

Without repeating all the points listed below, some of the most fundamental issues are the following:

- Reviewer #2 (point #1) suggests jointly modeling the effect of multiple features to examine potential feature interdependencies.
- Reviewer #2 (point #2) recommends including further analyses to examine potential mechanisms. Such analyses would significantly enhance the impact of the study and we would therefore strongly encourage you to perform them.
- Reviewers #1 and #3 provide several constructive suggestions on improving the performed analyses and strengthening the conclusions of the study.

All other issues raised by the reviewers need to be satisfactorily addressed.

REFeree REPORTS

Reviewer #1:

In this manuscript the authors continue previous work from the Landry lab analyzing the functional relationships between paralogs. In a previous study, they showed that some paralogous genes in yeast are functionally dependent rather than functionally redundant. Here, they extend this work to

human paralogs, and test whether the protection that paralogous genes provide against the effect of loss-of-function mutations would be dependent on whether their products form heteromeric complexes with each other. Computational analysis is based on 3,842 known, well-curated pairs of paralogs, and relies heavily on existing, large-scale data from CRISPR-Cas9 screens measuring the impact of gene loss-of-function on proliferation and mRNA abundance of cancer cell lines and one primary cell line. While the manuscripts reads well and the topic is of interest, I find the analyses problematic from various angles, as detailed below.

CS values correction and correlations: CS values are at the heart of the entire analysis. Yet, there were several issues with their usage and interpretation.

1. Page 6 lines 23-25: "CS values among datasets are correlated, indicating reproducible measurements of fitness effects across platforms, methodologies, cell lines and cell types ". However, looking at Fig. S2, apart from the correlation between the dependent datasets CS2 and CS2.1, correlations are not strong. In particular, the correlations with the only primary cell line, CS3, are low (0.18, 0.17). Being a primary cell line, CS3 supposedly contains less aberrations relative to the cancer cell lines used in the other CS screens, and therefore seems most reliable. The same statement about strong correlations appears in the Methods. The authors should accurately describe the correlations in the text and not mislead the reader.
2. In the various comparisons that rely on mean, use median instead (since mean is more sensitive to outliers).
3. The CS score is "The log₂ fold change between final read counts and initial read counts". Thus, a CS score of 0 means no change in growth. Looking at Figure 1, in all datasets except CS3, it appears that the mean behavior across all gene sets is that the mutants actually grow better than the wildtype. What is the explanation for this behavior?
4. Boxplots (as in Fig 1a) do not show the actual data points. The authors should present data points or make their distribution visible, for example by using beeswarm or violin plots. The median and/or quartiles can be marked on top.
5. 1B and 1C panels seem reversed (singletons shown in C, not in B, and vice versa).
6. Fig 1D - change mean to median; the trend across time, especially of the mean score, is hard to see/judge. What is the correlation? Looking at Fig. S4, it seems that the trend does not exist in any of the datasets, making this a rather untrustworthy result.

"We therefore first examined whether paralogous heteromers in humans likely derive from ancestral homomers and found that they indeed do, consistent with previous observations (Wagner 2003; Pereira-Leal et al. 2007)." -the authors should explicitly describe their analysis (datasets, numbers, etc), not just the conclusion.

Heteromeric paralogs: Any PPI is characterized as a heteromer/homomer, which is a terminology used for protein complexes, though it could be transient, indirect, etc. This seems like an over-stretch of the terminology.

The authors divided PPIs into all/ direct PPIs. Direct PPIs are potentially more reliable for this analysis. However, in Fig S5, showing direct PPIs, the trend is not statistically significant. The authors could try to repeat while using median instead of means, or at least state in the main text that it is insignificant.

Fig. 2B,C, Figs S5, S6 also show that most data points have mean CS score values > 0. Why then call the effects deleterious? Given that the numbers are positive, they may be less positive (see comment 3 above) but not deleterious, and not more negative. Fig. S5 - B-E the x axis label is hidden.

Fig. S6A - "Paralogs that form heteromers tend to be more deleterious upon loss-of-function than other paralogs". However in the 3 bins with the smallest values (leftmost), heteromeric paralogs are significantly more positive than the non-heteromeric ones. While the authors refer to this inconsistency in the main text, in the legend of the figure it is not mentioned at all. The inconsistency should be elaborated there too.

GO analysis: GO annotation can stem from various types of evidence. Which evidence codes were used? Annotations that are based on evidence such as inferred from homology, inferred from interactions, etc, should be removed, as they bias the analysis.

Figure 6: "Asymmetry of expression levels between paralogs creates an asymmetry in their functional importance" - the two features are correlates; the stated causality needs to be proven.

Minor:

In some places writing should be made clearer. For example, "Although this mutational robustness does not provide an advantage strong enough by itself to cause the maintenance of paralogs by natural selection-unless mutation rate or population size are exceptionally large (van Nimwegen, Crutchfield, and Huynen 1999)-paralogous genes nevertheless affect how biological systems globally respond to loss-of-function mutations"; consider dividing the sentence and try to avoid double negation.

Page 3 lines 38-39 partly repetitive.

gRNAs acronym: spell out.

Fig. S1 - the figure is not intuitive, a table might be a better option. S2B legend - "I 455 cell lines", remove the I.

Reviewer #2:

Dandage and colleagues study in this work how the properties of paralog genes is associated with differences in the impact of loss-of-function genetic perturbations. To perform this study the authors take advantage of a growing compilation of publicly available CRISPR based gene essentiality scores for a large panel of cancer cell lines. The main idea that is tested is whether the capacity of paralogs to compensate each other depends on them heteromerizing. While the authors find differences in the impact of gene deletion based on paralog (vs singletons) and heteromerization status, these differences are also confounded with the expression, protein interaction and biological processes of the genes involved. The main conclusion of this study appears to be that the paralogs that heteromerize are more essential than paralogs that don't heteromerize but it is unclear whether this is due to the interaction itself or if these other characteristics fully explain the differences. Overall, I think this is an interesting and well balanced study that advances our knowledge of the characteristics that determine the impact of loss-of-function mutations. I think the authors could attempt to provide a more elegant joint model of the characteristics and if possible further study some aspects related to mechanisms.

Major concerns

As the authors have shown, there are many characteristics that have an impact on the gene deletion scores - gene expression, number of interactions, biological process, paralog status, age, etc. These properties in turn as also quite interdependent. For example, protein interactions and gene expression levels can be inter-dependent. In the manuscript the authors often subset a feature (e.g. gene expression) in bins to test its impact on another metric such as the CS scores. A more elegant approach would have been to jointly model the impact of multiple variables in order to access how significant is the contribution of a given feature to the essentiality scores. Although there are multiple ways to perform such joint modelling one possibility would be to use a mixed linear model approach and comparing models using a likelihood ratio (LR) test. Specifically, the authors could build a model to predict the CS scores using the gene expression, protein interaction, age or any other gene feature they wish. A second model could be built with the addition of the paralog status for example and the two models compared with a LR test. This would more directly ask if paralog heteromerization has an impact when all other properties are jointly considered. The beta values of the different features in the model would also give an indication of the strength of association of each feature.

While the work is already extensive, it would have been useful to have additional work done on potential mechanisms. The authors mention some scenarios whereby the heteromerization may cause a decreased robustness. One scenario would be that the interaction stabilizes the proteins and the removal of one paralog could lead to a destabilization of the other. This would cause an overall trend of heteromerization associated with decreased robustness to LoF variants. If this would be the case then changes in expression levels of one paralog should associate with increased essentiality of

the other. Given the large number of samples used here, the authors could stratify the samples to test such hypothesis. For paralogs that heteromerize does the expression of one paralog relate with the essentiality of the other? Given that the gene expression of paralogs can correlate, the authors need to consider here the expression of the tested gene as a confounder in the analysis.

Related to the point above, the idea of interaction dependent protein interactions has been explored in Goncalves et al (<https://doi.org/10.1016/j.cels.2017.08.013>) and Sousa et al. (<https://www.biorxiv.org/content/10.1101/499434v1>). The latter containing a list of pairs of interacting proteins where such interaction mediated stability effect is detected albeit indirectly. Although the approach described in these manuscripts is only a predicted effect, the authors could study specific cases of paralogs that heteromerize where the interaction has a positive impact on the stability (and a decreased degradation) of the other.

Reviewer #3:

Dandage and Landry investigate whether the physical interactions between paralogs that heteromerize limit their ability to buffer each other's loss of function. While there is a vast amount of literature on the buffering of deleterious mutations by duplicated genes, the role of physical and functional interactions between paralogs in this process and the generality of the underlying mechanisms remain relatively unexplored. I believe that this work is of general interest and a valuable contribution in this direction. The analysis is based on the effects of gene inactivation on cell proliferation in human cancer cell lines and primary T-cells. This model system is particularly interesting, as understanding the mechanisms of functional compensation by duplicates in this setting could be relevant for cancer therapies.

The authors compared the effects (CS values) of gene loss-of-function on cell proliferation for genes stratified according to several variables. The main conclusion of the paper is that, on average, paralogs that form heteromers are more deleterious when inactivated than those that do not. Nevertheless, it is pointed out that this difference is likely a result of several confounding factors. Specifically, gene functions, mRNA expression, and the number of protein-protein interaction partners are discussed as the primary reasons for the stronger deleterious effects for heteromerizing paralogs. If true, this conclusion is highly significant, as it suggests that the physical dependence between paralogs is not a general causal mechanism for decreased functional compensation; a possibility that was raised by previous work in yeast by the same group (Diss et al. 2017 *Science* 355:630).

While the authors present a comprehensive analysis of an interesting question, I think there needs to be a better quantification of effect sizes to support the main conclusions. Without additional quantification it is not clear whether the difference between heteromeric and non-heteromeric paralogs can be attributed entirely to gene function, expression or PPI differences, or if there is an independent role of paralog physical interactions on buffering.

Major points

1. The authors conclude that heteromerizing paralogs are less protective than non-heteromeric paralogs largely due to their higher abundance and number of interaction partners. This conclusion is based on the binning of mRNA expression and PPI data shown in figure 4. It is not clear how the bin sizes were chosen, why are they not uniform, and whether they have an impact on the conclusions. Is it possible that the non-significant differences in CS values are simply a result of low statistical power due to a reduced number of paralogs in each bin? More importantly, after controlling for mRNA expression and number of interactions, is there still a fraction of the variance in CS values that can be explained by the physical interaction between paralogs? This is a key question that directly addresses the main hypothesis of the manuscript. The authors conclude that such a fraction is relatively minor based on the binned data, however it would be desirable to present a bin-free test for this conclusion.

One suggestion, which may address this question without binning the data, is to create a binary variable describing whether a paralog is heteromeric or non-heteromeric. The authors could then

look at the partial correlation of this variable with the CS values after controlling for either mRNA expression or the number of PPI interactions to quantify any independent contribution.

2. The authors demonstrate in figure 1 that singletons have lower CS values than duplicates. However, they then show in figure 4 and figure 5 that singletons have higher average expression levels than duplicates. I think it is critical to discuss how much of the difference in CS values between singletons and duplicates is explained by expression levels. This is relevant because without a clear difference between singletons and duplicates at the same level of expression, the comparison between heteromeric and non-heteromeric paralogs may not be interpreted in terms of buffering capacity. This can be addressed for example by plotting the CS values of singletons alongside duplicates at a given expression level, similar to figure 4D, or by doing a partial correlation analysis like the one described above.

3. In their analysis of molecular functions (figure 3), the authors show that GO terms enriched in heteromeric paralogs tend to have lower average CS values. It should be clarified whether the mean CS values per molecular function were calculated across all genes annotated with the GO term, or only duplicates; this was not clear from the text or figure legends. The authors suggest in the discussion that enrichment of certain molecular functions may explain the lower buffering capacity of heteromeric paralogs, but this is not tested directly.

To more directly address the hypothesis that heteromeric paralogs are less protective, the authors could compare, for each independent GO term, the mean CS values for heteromeric paralogs, non-heteromeric paralogs, and singletons. Paired tests across independent GO terms could show whether heteromerization plays a role in buffering mutations regardless of specific molecular functions, or whether heteromeric paralogs indeed behave more like singletons than duplicates, as proposed in the introduction. This may not be possible for all GO terms, but it should be doable for those with sufficient genes in each category.

4. While there seems to be a negative correlation between the age of paralogs and CS scores in figure 1D, this relationship is not quantified, furthermore such a pattern is much less evident in figure S4. I would encourage the authors to provide some measure of the significance of this trend either on the figure legends or the main text. For example, the coefficients and corresponding p-values for the correlation between the CS value of a duplicate pair and its dS.

Minor points

1. In figure 5 and figure S12 the contour lines showing the kernel density estimates for singletons, paralogs, heteromers, and non-heteromers have different scales. For example the outermost line for singletons in figure 5B corresponds to a density of 0.04, whereas the outermost line for paralogs corresponds to 0.016. This makes it hard to visualize whether two gene categories occupy different regions of the robustness landscape. I suggest using the same scales for the subsets of genes compared in each panel.

2. While it is claimed that CS values among experimental datasets are correlated (page 6), the correlation between CS3 and the other datasets is actually very weak ($r=0.18$). In the interest of transparency, I think the range of correlation coefficient values should be presented in the main text in addition to figure S2.

2. Check the legend of figure S6, the descriptions of panels B and C appear to be inverted.

3. Check the legend of figure S11, the descriptions of panels A and B appear to be inverted.

4. Figure S14 should be labeled figure S13.

Responses to reviewers comments

Responses are highlighted in blue. Text and figures reproduced from the manuscript are shown in boxes. Page numbers and line numbers correspond to the pdf version of manuscript file enclosed alongside.

Reviewer #1:

In this manuscript the authors continue previous work from the Landry lab analyzing the functional relationships between paralogs. In a previous study, they showed that some paralogous genes in yeast are functionally dependent rather than functionally redundant. Here, they extend this work to human paralogs, and test whether the protection that paralogous genes provide against the effect of loss-of-function mutations would be dependent on whether their products form heteromeric complexes with each other. Computational analysis is based on 3,842 known, well-curated pairs of paralogs, and relies heavily on existing, large-scale data from CRISPR-Cas9 screens measuring the impact of gene loss-of-function on proliferation and mRNA abundance of cancer cell lines and one primary cell line. While the manuscripts reads well and the topic is of interest, I find the analyses problematic from various angles, as detailed below.

CS values correction and correlations: CS values are at the heart of the entire analysis. Yet, there were several issues with their usage and interpretation.

1. Page 6 lines 23-25: "CS values among datasets are correlated, indicating reproducible measurements of fitness effects across platforms, methodologies, cell lines and cell types ". However, looking at Fig. S2, apart from the correlation between the dependent datasets CS2 and CS2.1, correlations are not strong. In particular, the correlations with the only primary cell line, CS3, are low (0.18, 0.17). Being a primary cell line, CS3 supposedly contains less aberrations relative to the cancer cell lines used in the other CS screens, and therefore seems most reliable. The same statement about strong correlations appears in the Methods.

The authors should accurately describe the correlations in the text and not mislead the reader.

We thank the reviewer for these suggestions. Considering the differences between the platforms, methodologies, cell lines and cell types in case of dataset CS1 and CS2 (and closely related dataset CS2.1), we would consider the correlations ($p=0.52$ in case of dataset CS2 and $p=0.51$ in case of CS2.1) to be appreciably high.

However, in case of dataset CS3, indeed, as the reviewer rightly pointed out, the correlations are low. Therefore, the relevant text is removed from the Methods section. The reason for this anomaly can indeed be attributed to the less aberrations of primary cells would have compared to the cancer cells. Thus CS3 may seem the most reliable dataset, however, CS3 is derived from a single study and experiment, it could have other idiosyncrasies that we cannot identify for the moment. Also, for human cells, it would be difficult to find a better dataset to test our hypothesis than the dataset we have now. Therefore, we think that our study of gene duplication in this context remains highly valuable. We hope that in the future more data on the primary cell lines would be

available. Taking this suggestion into account, we have given CS3 dataset an equal weightage as that of the other datasets in the metaanalysis.

The problematic sentence that the reviewer is referring to is now rephrased in the Results section (now on page 6 line no 6-11).

CS values among datasets CS1 and CS2/CS2.1 are well correlated, indicating reproducible measurements of fitness effects across platforms, methodologies, cell lines and cell types (Appendix Fig S1). The weaker correlation with dataset CS3 values (Spearman correlation coefficient ranges from 0.19 to 0.21), however could be attributed to the difference in the physiology of the primary and cancer cell lines itself, although technical factors could also be responsible.

In the discussion section, we also mention this as a limiting factor in our study (page 24 line 15-23).

Although the signal for the contribution of gene duplicates to robustness is significant and reproducible across datasets, some factors could limit the effects measured. The type of the cell lines used, i.e. cancer cell lines (in case of CS1, CS2/2.1) and primary (in case of CS3), clearly shows a difference in terms of the correlations (Appendix Fig S1). A second factor that is particular to the LOF screens in mammalian cell lines is the robustness of these cells to gene LOF. As seen from the distributions of the CS values across CS datasets, most of the genes are robust to LOF (Fig EV1). The effective range of deleteriousness is thereby very narrow, allowing the assessment of deleteriousness on a relative basis, rather than dependent on the absolute scores.

2. In the various comparisons that rely on mean, use median instead (since mean is more sensitive to outliers).

In the early stages of the study we had estimated both the mean and median scores, and we found that both of them correlate very strongly (as shown in Fig R1 below). Therefore, resultantly, all the trend we observe remain the same. However, indeed as rightly pointed out by the reviewer, mean is more sensitive to the outliers. What we rationalised is that mean would be more capture the shape of the distributions, for instance extremely deleterious gene inactivations-which is where the most significant signal may come from-more effectively than the median. Therefore, we carried our analysis with the mean scores. In order to allow readers to assess this, we have included both the mean and median scores in the source data (Dataset EV3) of the manuscript.

Fig R1: Correlations between the CS values calculated as means (y-axis) and medians (x-axis), in case gene-wise (left column) aggregation of the CS values and paralog-wise aggregation (right column).

Red line denotes the lowest regression. r: Pearson's correlation coefficient, P: P-value associated with the Pearson's correlation coefficient.

3. The CS score is "The log₂ fold change between final read counts and initial read counts". Thus, a CS score of 0 means no change in growth. Looking at Figure 1, in all datasets except CS3, it appears that the mean behavior across all gene sets is that the mutants actually grow better than the wildtype. What is the explanation for this behavior?

We thank the reviewer for pointing this issue out. This seems to be a peculiar feature of the loss of function screening in human cells. There are two aspects to this issue.

1. Most genes are only partially deleterious upon deletion.

Interestingly, across all CS datasets, it seems that most of the genes are only partially deleterious upon loss of function. Therefore the CS values of the majority of the genes are around 0, with quite a large dispersion around this value. We assume that this is partly due to the inherent robustness of the gene regulatory network in human cells and partly because functional importance of genes is context dependent, therefore not all genes contribute to the viability of the cells (i.e. the read-out) in the growth conditions of the co-culture competition assays. Some highly positive values could also represent beneficial effects on gene proliferation in the particular conditions assessed, for instance if the equivalent of tumor suppressor genes were inactivated.

2. The deleteriousness need to be assessed on a relative basis.

It is possible that the way the experiments are performed can only be interpreted in a relative fashion. Changes in cell clone abundance is estimated by counting relative reads in libraries and since the total number of reads is finite, even if all guides were given negative values, the use of relative read counts and normalization within a library would still produce positive changes. We therefore believe that these scores can only be largely interpreted in relative terms, unless internal controls are used to estimate absolute changes in read counts.

In order to address this issue, we have rescaled the CS data using a method that uses gold standard sets of essential and non-essential genes (Hart and Moffat 2016). In addition, these distributions clearly show that the way the nature of the data and the read normalization places the mode of the distributions above 0, which is most likely a statistical effect. One would expect the mode to be around 0. We therefore recommend that the CS values should be considered as relative values. We also show the distributions of the CS values in all 4 CS datasets with CS values of the sets of essential and non-essential genes marked on the distributions (see Fig EV1, also reproduced below). The locations of the essential and non-essential genes is clearly distinguishable.

Additionally, in the case of cancer cell lines, the CS values seem largely independent of the role of the genes in cancerogenesis and thereby aberrations that reviewer is referring to.

Fig EV1: Distribution of CS values in the 4 CS datasets.

The locations of the essential and non-essential genes (taken as a union set of genes reported by DepMap (DepMap 2018) and BAGEL (Hart and Moffat 2016)) are denoted on the distributions. Similarly, the locations of the cancer drivers, oncogenes and tumor suppressors are denoted on the distribution (derived from (Lever et al. 2019)).

4. Boxplots (as in Fig 1a) do not show the actual data points. The authors should present data points or make their distribution visible, for example by using beeswarm or violin plots. The median and/or quartiles can be marked on top.

We thank the reviewer for this suggestion. We have now changed all the boxplots to violin plots. Additionally, because violin plots shown in the figures do not cover the whole range of the data, data underlying all such plots is now included as Extended View datasets.

5. 1B and 1C panels seem reversed (singletons shown in C, not in B, and vice versa).

The correction has been carried out. Thanks to the reviewer for pointing it out.

6. Fig 1D - change mean to median; the trend across time, especially of the mean score, is hard to see/judge. What is the correlation? Looking at Fig. S4, it seems that the trend does not exist in any of the datasets, making this a rather untrustworthy result.

Our earlier approach of analysis may have been problematic, resulting in the signal from the data which was rather weak. In the revised manuscript, we revisited this analysis. We now use age of paralogs in terms of the evolutionary distance (expressed in terms of taxonomic levels, Fig 1D reproduced below).

This new approach additionally also provides stronger support for the analysis. However, we acknowledge that the signal remains rather weak and that age itself does not seem to be a major contributor (page 6 line 30-33).

Fig 1

D) Older paralogs tend to be more essential than younger one, therefore less protective (i.e. more deleterious upon LOF), than younger ones. On the y-axis, the age groups are ordered in increasing distance of phylogenetic node of duplication relative to common ancestor, i.e. Opisthokonta. Sets of essential and non-essential genes were derived from the union of gene sets reported by DepMap (DepMap 2018) and BAGEL (Hart and Moffat 2016) (See Methods). P-value from a two-sided Mann-Whitney U test is shown.

[7.] "We therefore first examined whether paralogous heteromers in humans likely derive from ancestral homomers and found that they indeed do, consistent with previous observations (Wagner 2003; Pereira-Leal et al. 2007)." -the authors should explicitly describe their analysis (datasets, numbers, etc), not just the conclusion.

In this case, the conclusion of the analysis actually preceded the analysis itself. In the revised manuscript, we have corrected the flow of the text in the section that the result.

Heteromeric paralogs emerge from ancestral homomers

The model in which paralogous genes are dependent on each other considers that interacting paralogs derive from ancestral homomeric proteins (Kaltenegger and Ober 2015; Baker, Hanson-Smith, and Johnson 2013; Bridgham et al. 2008; Diss et al. 2017). We can assume that when the two paralogs individually form a homomer, the ancestral protein was most likely also a homomer. Therefore, we can infer that heteromers of paralogs are derived from ancestral homomers, if each paralog also forms a homomer.

We use two sources of PPI (Livstone et al. 2011; Chatr-Aryamontri et al. 2015) and IntAct (Orchard et al. 2014), to define homomeric genes or heteromeric gene pairs based on PPI (see Methods). Further, the subsets are defined based on all PPI (henceforth this dataset will be referred to as 'all PPI') or direct physical interactions only (henceforth this dataset will be referred to as 'direct PPI'). Homomers, in the context of this study, refer to the assembly of a protein with itself while heteromers refer to paralogous proteins that assemble with other proteins. Considering all PPIs (see methods for the difference between 'all PPI' and 'direct PPI'), paralogs are 8.13 times more likely to form heteromeric pairs (Fisher's exact test, P-value < 1.4e-14) if they also both form homomers than if none of them does. The likelihood is 48.88 times for heteromers defined by 'direct PPI's only (P-value < 5.5e-18) (see Appendix Table S1 for the numbers of pairs in each category). We can therefore generally assume that pairs of heteromers are more likely to derive from ancestral homomers, consistent with previous observations (Wagner 2003; Pereira-Leal et al. 2007).

[8.] Heteromeric paralogs: Any PPI is characterized as a heteromer/homomer, which is a terminology used for protein complexes, though it could be transient, indirect, etc. This seems like an over-stretch of the terminology.

We have used this term in other manuscripts but we do agree with the reviewer that this can be an issue, so we thank the reviewer for pointing that out. Since homomer refers to complexes that are formed by a single subunit and heteromeric refers to those that have more than one component, referring to heteromeric paralogs does only imply that these paralogs form complexes with more than one subunit. We have not found any clear alternative way of designating these complexes. We therefore precisely define what we mean early in the paper (page 8 line 9-15) to make sure there is no confusion.

[9.] The authors divided PPIs into all/ direct PPIs. Direct PPIs are potentially more reliable for this analysis. However, in Fig S5, showing direct PPIs, the trend is not statistically significant. The authors could try to repeat while using median instead of means, or at least state in the main text that it is insignificant.

This is a very useful suggestion. With the reanalysed CS scores, we do find significant differences in case of direct interactions, for all CS datasets except CS3 dataset (now Appendix Fig S3). We think that this may be partly due to the addition PPI data from IntAct database (Orchard et al. 2014) that cover a large number of PPIs collected by

yeast-2-hybrid (Luck et al. 2019). Also, we produced similar plots with the median analysed CS scores and find that it is very much similar to the one with mean scores (see Fig R2 below). This follows the response to the point #2. The mean analysed CS values are very strongly correlated with the median analysed ones. Finally, we agree that direct PPIs may be more reliable but because they remain a relatively smaller set, it offers a lower statistical power. We therefore systematically present results with both PPI datasets (analysis with 'all PPI' in Fig 2A and analysis with 'direct PPI' in Appendix Fig S3) and overall the results seem to agree with each other.

Fig R2: Comparison between the mean analysed (left column) and median analysed (right column) CS values.

P-values of two-sided Mann-Whitney U tests are shown.

[10.] Fig. 2B,C, Figs S5, S6 also show that most data points have mean CS score values > 0 . Why then call the effects deleterious? Given that the numbers are positive, they may be less positive (see comment 3 above) but not deleterious, and not more negative. Fig. S5 - B-E the x axis label is hidden.

This is point is related to the earlier comment #3. Due to the outlined issues, we recommend that the deleteriousness of the genes should be assessed on relative terms.

While generating the figures we have made sure that the axis labels are visible. The plots reviewer is referring to are now located at Appendix Fig S4.

[11.] Fig. S6A - "Paralogs that form heteromers tend to be more deleterious upon loss-3 of-function than other paralogs". However in the 3 bins with the smallest values (leftmost), heteromeric paralogs are significantly more positive than the non-heteromeric ones. While the authors refer to this inconsistency in the main text, in the legend of the figure it is not mentioned at all. The inconsistency should be elaborated there too.

The analyses with dS values were inconsistent at other places too (eg. previous Figure 1D). Therefore in the rewritten manuscript we rely on the age groups (in terms of taxonomy level) that were retrieved from Ensembl Compara database (Herrero et al.

2016).

With this new approach too, we see the inconsistency which we acknowledge the in the figure legends (Fig 2E) and also in the discussion (page 25 line 25-27).

[12.] GO analysis: GO annotation can stem from various types of evidence. Which evidence codes were used? Annotations that are based on evidence such as inferred from homology, inferred from interactions, etc, should be removed, as they bias the analysis.

This is an important point. In the newer version of the manuscript, the GO analysis is redone, implementing the reviewer's suggestions. The gene set annotations are provided as Dataset EV4. It is extremely difficult to determine what is the source of the annotations in the Gene Ontology analyses, but we believe they are unlikely to be based on the large-scale experimental data we are using to assign paralogs to heteromeric and non heteromeric ones. Also, we think that the biases caused by the annotations based on protein sequences would most likely affect all paralogs, irrespective of their heteromeric status.

Finally, we acknowledge in the method section that GO gene sets may originate from evidence which may not entirely be independent from the rest of the data used in the meta-analysis (page 30 line 41-43).

Note that GO gene sets may originate from evidences which may not entirely be independent from the rest of the data used in the metaanalysis.

Also, In compliance with the reviewer's request, we have now removed evidence based on sequence homology (Inferred from Sequence Orthology (ISO)) and protein interactions (Inferred from Physical Interaction (IPI)) (see Methods section **GO enrichment analysis**). This modification in the methods however, only slightly change our previous results. We find that heteromer are enriched for gene sets containing proteins that have catalytic activity and known to directly interact/regulate with each other such as kinase binding as well as DNA binding proteins from histone deacetylase binding gene set (Results of the GO term enrichment analysis are provided as Dataset EV5).

[13.] Figure 6: "Asymmetry of expression levels between paralogs creates an asymmetry in their functional importance" - the two features are correlates; the stated causality needs to be proven.

We thank the reviewer for pointing it out because we think this is one of the most interesting observations of our manuscript and it relates to other points raised by other reviewers. We do not think the role of expression alone in determining which paralog is most important for fitness can be proven from data analysis alone. Additional experiments would be needed to show that. We are currently performing such experiments in yeast. However, our most important result here is that this asymmetry

seems to be more impactful for heteromeric paralogs. We now provide more discussion about this aspect (as discussed in response to reviewer#2's comment #2).

Minor:

In some places writing should be made clearer. For example, "Although this mutational robustness does not provide an advantage strong enough by itself to cause the maintenance of paralogs by natural selection-unless mutation rate or population size are exceptionally large (van Nimwegen, Crutchfield, and Huynen 1999)-paralogous genes nevertheless affect how biological systems globally respond to loss-of-function mutations"; consider dividing the sentence and try to avoid double negation.

The sentence in concern is now divided into two. The correction has been made on page 3 line 6-9. The text is also reproduced below.

This mutational robustness does not provide an advantage strong enough by itself to cause the maintenance of paralogs by natural selection unless mutation rate or population size are exceptionally large (van Nimwegen, Crutchfield, and Huynen 1999). Nevertheless, paralogous genes affect how biological systems globally respond to loss-of-function (LOF) mutations.

Page 3 lines 38-39 partly repetitive.

We thank the reviewer for pointing this out. The suggested correction has been made at page 3 line 37-39. The relevant text is reproduced below.

This would improve our understanding of evolution and also accelerate the development of medical interventions because redundancy is often a major obstacle in this context (Lavi 2015).

gRNAs acronym: spell out.

The suggested correction has been made on page 5 at line 36.

Fig. S1 - the figure is not intuitive, a table might be a better option. S2B legend - "1 455 cell lines", remove the 1.

The suggested correction has been made, instead of the figure, we have included a table as suggested by the reviewer: Table EV1.

Reviewer #2:

Dandage and colleagues study in this work how the properties of paralog genes is associated with differences in the impact of loss-of-function genetic perturbations. To perform this study the authors take advantage of a growing compilation of publicly available CRISPR based gene essentiality scores for a large panel of cancer cell lines. The main idea that is tested is whether the capacity of paralogs to compensate each other depends on them heteromerizing. While the authors find differences in the impact of gene deletion based on paralog (vs singletons) and heteromerization status, these differences are also confounded with the expression, protein interaction and biological processes of the genes involved. The main conclusion of this study appears to be that the paralogs that heteromerize are more essential than paralogs that don't heteromerize but it is unclear whether this is due to the interaction itself or if these other characteristics fully explain the differences. Overall, I think this is an interesting and well balanced study that advances our knowledge of the characteristics that determine the impact of loss-of-function mutations. I think the authors could attempt to provide a more elegant joint model of the characteristics and if possible further study some aspects related to mechanisms.

We thank the reviewer for this very positive review and for his appreciation of the importance of this type of work.

Major concerns

[1] As the authors have shown, there are many characteristics that have an impact on the gene deletion scores - gene expression, number of interactions, biological process, paralog status, age, etc. These properties in turn are also quite interdependent. For example, protein interactions and gene expression levels can be inter-dependent. In the manuscript the authors often subset a feature (e.g. gene expression) in bins to test its impact on another metric such as the CS scores. A more elegant approach would have been to jointly model the impact of multiple variables in order to assess how significant is the contribution of a given feature to the essentiality scores. Although there are multiple ways to perform such joint modelling one possibility would be to use a mixed linear model approach and comparing models using a likelihood ratio (LR) test. Specifically, the authors could build a model to predict the CS scores using the gene expression, protein interaction, age or any other gene feature they wish. A second model could be built with the addition of the paralog status for example and the two models compared with a LR test. This would more directly ask if paralog heteromerization has an impact when all other properties are jointly considered. The beta values of the different features in the model would also give an indication of the strength of association of each feature.

This is truly one of the issues that that we expressed our concern in earlier version of the manuscript. We thank the reviewer for the suggestions joint modeling of multiple features.

Among the choices of approaches to jointly model the interdependent features, we rather make use of partial correlations and ensemble machine learning models. Partial

correlation approach, in particular, allows us to assess the deleteriousness of heteromerization while jointly considering mRNA expression and number of PPI per gene (as shown in Fig 4D). As seen in this analysis, the correlation between a paralog status i.e. heteromer or not (binary variable, 1: heteromer, 0: not heteromer) with CS values (across 4 CS datasets) is almost lost when controlling for both mRNA expression (denoted as 'expression') and number of PPI partners (denoted as 'interactions'), indicating that the interdependency of the features is playing a significant part in this context.

Partial correlations also provide proxies for the strength of each of the molecular features. From Fig 4D, it is also apparent that when controlled for number of interactions, the correlation is lost more than if controlled for expression alone, suggesting that the number of interactions is a stronger determinant of the correlation and thus the deleteriousness of the heteromers.

As an independent approach, we also estimate the feature importance using ensemble modeling using 4 machine learning models (shown in Fig 4E). Corroborating with the results of the partial correlation analysis, it also shows that the number of interactions is a stronger predictor of deleteriousness of the heteromers than mRNA expression.

Given that the two adopted approaches served the purpose of jointly modelling the molecular features, we haven't attempted to use linear mixed models, as suggested by the reviewer. However, we appreciate this particular suggestion of method by the reviewer. In our response, we have taken the same general direction as advised by the reviewer, but using different approaches, which we believe effectively serves the purpose.

[2] While the work is already extensive, it would have been useful to have additional work done on potential mechanisms. The authors mention some scenarios whereby the heteromerization may cause a decreased robustness. One scenario would be that the interaction stabilizes the proteins and the removal of one paralog could lead to a destabilization of the other. This would cause an overall trend of heteromerization associated with decreased robustness to LoF variants. If this would be the case then changes in expression levels of one paralog should associate with increased essentiality of the other. Given the large number of samples used here, the authors could stratify the samples to test such hypothesis. For paralogs that heteromerize does the expression of one paralog relate with the essentiality of the other? Given that the gene expression of paralogs can correlate, the authors need to consider here the expression of the tested gene as a confounder in the analysis.

As rightly pointed out by the reviewer, from our analysis it appears that in the case of heteromers, the expression of one paralog is related to the essentiality of the other.

We tested if the difference in mRNA expression of highly expressed paralog (P1) with low expressed one (P2) is related to the essentiality difference between the two paralogs. Specifically, we correlated the normalised difference in the expression of paralogs $((P1-P2)/(P1+P2))$, where mRNA expression of P1 is greater than that of P2, also referred to as asymmetry of mRNA expression in the text), to the difference in the essentiality of the paralogs $(P1(CS)-P2(CS))$. These correlations were carried out for heteromeric as well as non-heteromeric paralogs across 374 cell lines. In case of the heteromers there seems to be a more negative correlation (Fig 6C and Fig EV5B). The negative correlation implies that if the paralogs symmetrically expressed, the difference between the deleteriousness of the paralogs upon LOF increases. This analysis shows that changes in expression of one paralog is related to essentiality of other paralog.

However, we would like to note that while replying to the reviewer's comment, we did not understand what the reviewer meant in the very last sentence of the comment. If possible, we would like some clarification on that.

[3] Related to the point above, the idea of interaction dependent protein interactions has been explored in Goncalves et al (<https://doi.org/10.1016/j.cels.2017.08.013>) and Sousa et al. (<https://www.biorxiv.org/content/10.1101/499434v1>). The latter containing a list of pairs of interacting proteins where such interaction mediated stability effect is detected albeit indirectly. Although the approach described in these manuscripts is only a predicted effect, the authors could study specific cases of paralogs that heteromerize where the interaction has a positive impact on the stability (and a decreased degradation) of the other.

We assume that the reviewer meant "interaction dependent protein expression" as found by Goncalves et al. (Gonçalves et al. 2017) and Sousa et al. (Sousa et al. 2019) rather than "interaction dependent protein interactions".

While working on the revision of the manuscript, we took the reviewer's comment (we assume) and looked into the dataset from Sausa et al study (Sousa et al. 2019). Because the reported attenuated set of genes were not found to show a significant overlap with heteromeric paralogs, we couldn't conclusively attach that aspect to our set of results. Nevertheless, inspired from that study, we correlated the deleteriousness of the heteromers with the interaction strength (size of interface used as a proxy). This analysis provided a deeper insight into the deleteriousness of the heteromeric paralogs. Therefore, overall, we believe that the suggested papers were beneficial in the revision

of the manuscript. We thank the reviewer for that.

Reviewer #3:

Dandage and Landry investigate whether the physical interactions between paralogs that heteromerize limit their ability to buffer each other's loss of function. While there is a vast amount of literature on the buffering of deleterious mutations by duplicated genes, the role of physical and functional interactions between paralogs in this process and the generality of the underlying mechanisms remain relatively unexplored. I believe that this work is of general interest and a valuable contribution in this direction. The analysis is based on the effects of gene inactivation on cell proliferation in human cancer cell lines and primary T-cells. This model system is particularly interesting, as understanding the mechanisms of functional compensation by duplicates in this setting could be relevant for cancer therapies.

The authors compared the effects (CS values) of gene loss-of-function on cell proliferation for genes stratified according to several variables. The main conclusion of the paper is that, on average, paralogs that form heteromers are more deleterious when inactivated than those that do not. Nevertheless, it is pointed out that this difference is likely a result of several confounding factors. Specifically, gene functions, mRNA expression, and the number of protein-protein interaction partners are discussed as the primary reasons for the stronger deleterious effects for heteromerizing paralogs. If true, this conclusion is highly significant, as it suggests that the physical dependence between paralogs is not a general causal mechanism for decreased functional compensation; a possibility that was raised by previous work in yeast by the same group (Diss et al. 2017 Science 355:630).

While the authors present a comprehensive analysis of an interesting question, I think there needs to be a better quantification of effect sizes to support the main conclusions. Without additional quantification it is not clear whether the difference between heteromeric and non-heteromeric paralogs can be attributed entirely to gene function, expression or PPI differences, or if there is an independent role of paralog physical interactions on buffering.

We thank the reviewer for considering our work as of general interest and a valuable contribution.

Major points

1. The authors conclude that heteromerizing paralogs are less protective than non-heteromeric paralogs largely due to their higher abundance and number of interaction partners. This conclusion is based on the binning of mRNA expression and PPI data shown in figure 4. It is not clear how the bin sizes were chosen, why are they not uniform, and whether they have an impact on the conclusions. Is it possible that the non-significant differences in CS values are simply a result of low statistical power due to a reduced number of paralogs in each bin? More importantly, after controlling for mRNA expression and number of interactions, is there still a fraction of the variance in CS values that can be explained by the physical interaction between paralogs? This is a key question that directly addresses the main hypothesis of the manuscript. The authors

conclude that such a fraction is relatively minor based on the binned data, however it would be desirable to present a bin-free test for this conclusion.

One suggestion, which may address this question without binning the data, is to create a binary variable describing whether a paralog is heteromeric or non-heteromeric. The authors could then look at the partial correlation of this variable with the CS values after controlling for either mRNA expression or the number of PPI interactions to quantify any independent contribution.

In order to address reviewer's comments on Fig 4, we have replaced the binning-based approach with partial correlation analysis as suggested by the reviewer. We thank the reviewer for this very useful comment. It definitely allows us to better establish conclusions of the relevant analyses.

As shown in Fig 4D, controlling for either expression (mRNA expression) or interactions (number of PPI partners) or both, diminishes the correlation between binary variable indicating whether the paralog is heteromer or not and CS scores when none of the factors were controlled for. This suggests that there is an interdependence among the features. Moreover, as the correlation diminishes more in case of interaction than expression, demonstrating that the deleteriousness of the heteromers is more dependent on the number of interactions.

Fig 4:

D) Partial Spearman correlation coefficients (r , shown on the y axis), between CS values and a paralog status (heteromer or not, binary variable, 1 : heteromer, 0 : not heteromer). The correlations were determined while controlling for none of mRNA expression and number of interactions (“none”), only mRNA expression (“expression”), only number of interactions (“interaction”) or both (“both”) (as shown on the x axis). Controlling for the number of interactions leads to the greater loss of negative correlation, indicating that it contributes to the correlation more than mRNA expression. Similar analysis with heteromers defined by ‘direct PPI’ is shown in Appendix Fig S8E.

2. The authors demonstrate in figure 1 that singletons have lower CS values than duplicates. However, they then show in figure 4 and figure 5 that singletons have higher average expression levels than duplicates. I think it is critical to discuss how much of the difference in CS values between singletons and duplicates is explained by expression levels. This is relevant because without a clear difference between singletons and duplicates at the same level of expression, the comparison between heteromeric and non-heteromeric paralogs may not be interpreted in terms of buffering capacity. This can be addressed for example by plotting the CS values of singletons alongside duplicates at a given expression level, similar to figure 4D, or by doing a partial correlation analysis like the one described above.

This is indeed an aspect that was underdeveloped in the earlier version of the manuscript. We agree with the reviewer that this aspect should have been discussed in more detail. We thank the reviewer for pointing this issue out and offering suggestions after implementing seem to improve this aspect of the manuscript.

As suggested by the reviewer, we have carried out the partial correlation analysis with the paralog or singleton status of the genes, similar to the one we carried out in response to the previous comment (comment #1). It shows that expression indeed is a major contributor in the deleteriousness of the duplicates versus singletons (shown in Fig EV3D). Further looking at the CS values at the range of expression values, the interdependence is clearly apparent (Fig EV3E). Only at the high expression values the CS values of paralogs are significantly different than that of the singletons. As discussed in the text (page 15 line 7-9), the reason for this interdependence could arise because small counts of expression scores are often more noisy.

3. In their analysis of molecular functions (figure 3), the authors show that GO terms enriched in heteromeric paralogs tend to have lower average CS values. It should be clarified whether the mean CS values per molecular function were calculated across all genes annotated with the GO term, or only duplicates; this was not clear from the text or figure legends. The authors suggest in the discussion that enrichment of certain molecular functions may explain the lower buffering capacity of heteromeric paralogs, but this is not tested directly.

To more directly address the hypothesis that heteromeric paralogs are less protective, the authors could compare, for each independent GO term, the mean CS values for heteromeric paralogs, non-heteromeric paralogs, and singletons. Paired tests across

independent GO terms could show whether heteromerization plays a role in buffering mutations regardless of specific molecular functions, or whether heteromeric paralogs indeed behave more like singletons than duplicates, as proposed in the introduction. This may not be possible for all GO terms, but it should be doable for those with sufficient genes in each category.

The mean CS values used in the GO term enrichment analysis belong to the paralogs (duplicates) only. We thank the reviewer for pointing this out. We now mention explicitly that the relevant analysis is done at the level of paralogs (on page 11 on line 23-24) and average CS value per paralog was used in the analysis (legend of the figure page 13 line 5-6). The enrichment of the terms is estimated by taking the total number of paralogs in account.

Regarding the speculation made in the Discussion, the molecular functions found to be enriched among heteromers and are on average significantly more deleterious than non-heteromers are associated with essential molecular functions related to transcription process, leading us to make that speculation. In our opinion the speculation in the concern, mentioned in the Discussion section is reasonable. Therefore we haven't made any change there. Also, the gene set enrichment analysis itself was the test that we performed that lead us to report that speculation.

In the revised manuscript, we implement the helpful suggestion of the reviewer and check for the mean CS values each independent GO term and monitor if the difference between the CS value of the heteromer and non-heteromer is significant (Fig 3). The gene sets with significant difference are indicated on the figure itself. In addition to molecular functions, we also test the biological processes, and cellular components. The GO analysis indicates the sets of gene sets that are enriched with heteromers and thus have lower buffering capacity. We believe that the analysis suggests that heteromerization and molecular function jointly play a role in the greater relative deleteriousness of the heteromers. Therefore, as a conclusion of this analysis, we report that the deleteriousness of the heteromers can be attributed to the molecular functions also.

4. While there seems to be a negative correlation between the age of paralogs and CS scores in figure 1D, this relationship is not quantified, furthermore such a pattern is much less evident in figure S4. I would encourage the authors to provide some measure of the significance of this trend either on the figure legends or the main text. For example, the coefficients and corresponding p-values for the correlation between the CS value of a duplicate pair and its dS.

This is an issue that was also pointed out by reviewer#1. In order to simplify our point, we now compare the age of the paralogs in terms of evolutionary distance (age groups obtained from Ensembl compara database (Herrero et al. 2016)) of the paralogous genes of essential and non-essential genes. This analysis shows that the older paralogs are more likely essential and hence more likely deleterious upon LOF.

Minor points

1. In figure 5 and figure S12 the contour lines showing the kernel density estimates for singletons, paralogs, heteromers, and non-heteromers have different scales. For example the outermost line for singletons in figure 5B corresponds to a density of 0.04, whereas the outermost line for paralogs corresponds to 0.016. This makes it hard to visualize whether two gene categories occupy different regions of the robustness

landscape. I suggest using the same scales for the subsets of genes compared in each panel.

The kernel density estimates are actually data-aware and we do not expect the densities of the genes of each gene sets to be similarly distributed. Therefore, it would not be possible to rescale the values. However, the shades of colors in each density contour are in same scale. Therefore the reader will be able to judge the relative levels of densities of the genes, regardless of the absolute values of the kernel density. Because of the above reasons, we haven't implemented this suggestion. Although, taking reviewer's in the account, in the revised version of the manuscript we have attempted to improve the readability of the figure, for instance, by using the full scale of expression and interaction values.

2. While it is claimed that CS values among experimental datasets are correlated (page 6), the correlation between CS3 and the other datasets is actually very weak ($r=0.18$). In the interest of transparency, I think the range of correlation coefficient values should be presented in the main text in addition to figure S2.

We have implemented this suggestion. We report the correlation coefficients and acknowledge the inconsistency in the main text on page 6 line 6-11.

CS values among datasets CS1 and CS2/CS2.1 are well correlated, indicating reproducible measurements of fitness effects across platforms, methodologies, cell lines and cell types (Appendix Fig S1). The weaker correlation with dataset CS3 values (Spearman correlation coefficient ranges from 0.19 to 0.21), however could be attributed to the difference in the physiology of the primary and cancer cell lines itself, although technical factors could also be responsible.

2. Check the legend of figure S6, the descriptions of panels B and C appear to be inverted.

The binning based analysis in the previous Figure S6 has been removed as per the suggestion of reviewer #1. Therefore this correction could not be implemented.

3. Check the legend of figure S11, the descriptions of panels A and B appear to be inverted.

The panels of the figure, now located at Appendix Fig S8, are now referred correctly in the revised version of the manuscript.

4. Figure S14 should be labeled figure S13.

Due to changes in the supplementary/Appendix figures, this correction could not be implemented.

References

- Baker, Christopher R., Victor Hanson-Smith, and Alexander D. Johnson. 2013. "Following Gene Duplication, Paralog Interference Constrains Transcriptional Circuit Evolution." *Science* 342 (6154): 104–8.
- Bridgham, Jamie T., Justine E. Brown, Adriana Rodríguez-Marí, Julian M. Catchen, and Joseph W. Thornton. 2008. "Evolution of a New Function by Degenerative Mutation in Cephalochordate Steroid Receptors." *PLoS Genetics* 4 (9): e1000191.
- Chatr-Aryamontri, Andrew, Bobby-Joe Breitkreutz, Rose Oughtred, Lorrie Boucher, Sven Heinicke, Daici Chen, Chris Stark, et al. 2015. "The BioGRID Interaction Database: 2015 Update." *Nucleic Acids Research* 43 (Database issue): D470–78.
- DepMap, Broad. 2018. "DepMap Achilles 18Q3 Public." Figshare. <https://doi.org/10.6084/M9.FIGSHARE.6931364.V1> [DATASET].
- Diss, Guillaume, Isabelle Gagnon-Arsenault, Anne-Marie Dion-Coté, Hélène Vignaud, Diana I. Ascencio, Caroline M. Berger, and Christian R. Landry. 2017. "Gene Duplication Can Impart Fragility, Not Robustness, in the Yeast Protein Interaction Network." *Science* 355 (6325): 630–34.
- Gonçalves, Emanuel, Athanassios Fragoulis, Luz Garcia-Alonso, Thorsten Cramer, Julio Saez-Rodriguez, and Pedro Beltrao. 2017. "Widespread Post-Transcriptional Attenuation of Genomic Copy-Number Variation in Cancer." *Cell Systems* 5 (4): 386–98.e4.
- Hart, Traver, and Jason Moffat. 2016. "BAGEL: A Computational Framework for Identifying Essential Genes from Pooled Library Screens." *BMC Bioinformatics*. <https://doi.org/10.1186/s12859-016-1015-8>.
- Herrero, Javier, Matthieu Muffato, Kathryn Beal, Stephen Fitzgerald, Leo Gordon, Miguel Pignatelli, Albert J. Vilella, et al. 2016. "Ensembl Comparative Genomics Resources." *Database: The Journal of Biological Databases and Curation* 2016 (May). <https://doi.org/10.1093/database/baw053>.
- Kaltenegger, Elisabeth, and Dietrich Ober. 2015. "Paralogue Interference Affects the Dynamics after Gene Duplication." *Trends in Plant Science* 20 (12): 814–21.
- Lavi, Orit. 2015. "Redundancy: A Critical Obstacle to Improving Cancer Therapy." *Cancer Research* 75 (5): 808–12.
- Lever, Jake, Eric Y. Zhao, Jasleen Grewal, Martin R. Jones, and Steven J. M. Jones. 2019. "CancerMine: A Literature-Mined Resource for Drivers, Oncogenes and Tumor Suppressors in Cancer." *Nature Methods* 16 (6): 505–7.
- Livstone, Michael, Michael Livstone, Bobby-Joe Breitkreutz, Chris Stark, Lorrie Boucher, Andrew Chatr-Aryamontri, Rose Oughtred, et al. 2011. "The BioGRID Interaction Database." *Nature Precedings*. <https://doi.org/10.1038/npre.2011.5627.1>.
- Luck, K., D. K. Kim, L. Lambourne, K. Spirohn, and B. E. Begg. 2019. "A Reference Map of the Human Protein Interactome." *bioRxiv*. <https://www.biorxiv.org/content/10.1101/605451v1.abstract>.
- Nimwegen, E. van, J. P. Crutchfield, and M. Huynen. 1999. "Neutral Evolution of Mutational Robustness." *Proceedings of the National Academy of Sciences* 96 (17): 9716–20.
- Orchard, Sandra, Mais Ammari, Bruno Aranda, Lionel Breuza, Leonardo Briganti, Fiona Broackes-Carter, Nancy H. Campbell, et al. 2014. "The MIntAct Project--IntAct as a Common Curation Platform for 11 Molecular Interaction Databases." *Nucleic Acids Research* 42 (Database issue): D358–63.
- Pereira-Leal, Jose B., Emmanuel D. Levy, Christel Kamp, and Sarah A. Teichmann. 2007. "Evolution of Protein Complexes by Duplication of Homomeric Interactions." *Genome Biology* 8 (4): R51.
- Sousa, Abel, Emanuel Gonçalves, Bogdan Mirauta, David Ochoa, Oliver Stegle, and Pedro

Beltrao. 2019. "Multi-Omics Characterization of Interaction-Mediated Control of Human Protein Abundance Levels." *Molecular & Cellular Proteomics: MCP*, June.
<https://doi.org/10.1074/mcp.RA118.001280>.

Wagner, Andreas. 2003. "How the Global Structure of Protein Interaction Networks Evolves." *Proceedings. Biological Sciences / The Royal Society* 270 (1514): 457–66.

Thank you again for sending us your revised manuscript. We have now heard back from the two referees who agreed to evaluate your study. As you will see below, the reviewers acknowledge that the study has significantly improved as a result of the performed revisions. However, reviewer #3 raises a few remaining concerns, which we would ask you to address in a minor revision.

REFEREE REPORTS

Reviewer #2:

The authors have addressed all my previous concerns - they provide now a join model of the features and performed some additional analysis of mechanisms.

Reviewer #3:

In this second iteration the authors have made a thorough effort to address the reviewers' concerns. I think the updated manuscript provides a better picture of how the considered properties of paralogs interact and relate to loss of function phenotypes. I think a little work would greatly help improve clarity in some parts of the manuscript.

Main comments:

1. On page 14, the authors have now included the results of partial correlation analyses and several machine learning models to predict paralog states. While these results are a valuable addition to this work, I recommend that the authors try to improve the clarity of the text. For example, on line 15 it says: "Considering the interdependence between the three related factors, partial correlation between each two factors was calculated while controlling for the third factor". However, the remaining text in that paragraph does not describe the partial correlation results, which are presented (in a somewhat convoluted way) in the next paragraph.

Perhaps the authors could separate the sentences describing 1) the differences in PPIs and mRNA levels of heteromeric paralogs, 2) the partial correlation analysis description and results, and 3) the machine learning analysis description and results.

2. While it is interesting that PPIs are more relevant to predict the heteromeric state of paralogs than expression or CS scores (Figure 4E). A more appropriate question to ask using the machine learning models was how heteromeric states compare to PPIs and expression in predicting CS scores. While this is addressed by the partial correlation analyses, it is not directly corroborated by the implemented models.

3. Page 15, line 6 : "In the case of all the CS datasets, at lower expression values, the difference in CS between paralogs and singletons is non-significant (Fig EV3E). The reason for this could be attributed to the small counts of mRNA expression generally being relatively more noisy". While expression measurements are indeed noisier at lower expression levels, it is not immediately clear how this would explain the similar CS values between paralogs and singletons. Specifically, because the values of these two variables (paralog vs. singleton, CS value) are calculated without using any expression measurements.

Minor points:

2. The order of labels in figure 2E could be re-arranged to reflect taxonomic hierarchy. In the figure legend (page 11, line 7) it says that paralogs are ordered by their age, but shouldn't then *Homo sapiens* be either at the top or bottom of the list?

3. According to the authors' reply, the X-axis of the left panel of figure 3 should say "CS mean (all paralogs)" instead of "CS mean (heteromer)". Also, it may be easier to read "proportion of heteromers" instead of "# of heteromers/ # of paralogs", in the figure.

Responses to reviewers' comments

Note that our responses are highlighted in black. Text and figures reproduced from the revised manuscript are shown in boxes. Page numbers and line numbers correspond to the pdf version of manuscript file enclosed alongside.

Reviewer #2:

The authors have addressed all my previous concerns - they provide now a joint model of the features and performed some additional analysis of mechanisms.

We greatly appreciate the overall insightful comments provided by the reviewer in the major review of our manuscript and especially the suggestion to incorporate the joint models and mechanistic analysis.

Reviewer #3:

In this second iteration the authors have made a thorough effort to address the reviewers' concerns. I think the updated manuscript provides a better picture of how the considered properties of paralogs interact and relate to loss of function phenotypes. I think a little work would greatly help improve clarity in some parts of the manuscript.

We thank the reviewer for the comprehensive assessment of our manuscript in the major review. The issues pointed out by the reviewer and corresponding suggestions have helped us to strengthen our arguments in the manuscript.

Main comments:

1. On page 14, the authors have now included the results of partial correlation analyses and several machine learning models to predict paralog states. While these results are a valuable addition to this work, I recommend that the authors try to improve the clarity of the text. For example, on line 15 it says: "Considering the interdependence between the three related factors, partial correlation between each two factors was calculated while controlling for the third factor". However, the remaining text in that paragraph does not describe the partial correlation results, which are presented (in a somewhat convoluted way) in the next paragraph.

Perhaps the authors could separate the sentences describing 1) the differences in PPIs and mRNA levels of heteromeric paralogs, 2) the partial correlation analysis description and results, and 3) the machine learning analysis description and results.

It is unfortunate that the way the results of partial correlation analysis were written appeared convoluted to the reviewer. In response to the reviewer's comment, we have now rewritten this section of the result to make it easier to follow.

Actually, there are two separate partial correlation analyses discussed in that section of the results. At the start of the section, we discuss about the analysis shown in Fig 4A while subsequently we discuss the results of the analysis shown in Fig 4D. Indeed a

clear distinction was needed there. In order to make it clear, following the suggestion of the reviewer, we have split the first paragraph. We also moved the sentence describing pointed out by the reviewer to the legends of Fig 4A (page 26 line 4 to 7). We reasoned that in the context of analysis of Fig 4A, use of partial correlation analysis is a methodological detail that is better be noted in the legend of the figure, in order to avoid any potential confusion.

Likewise, we have also separated the joint modelling analyses i.e. partial correlation analysis (shown in Fig 4D) and machine learning analysis (shown in Fig 4E) into separate paragraphs. We believe that the overall minor editing has improved the readability of this section.

2. While it is interesting that PPIs are more relevant to predict the heteromeric state of paralogs than expression or CS scores (Figure 4E). A more appropriate question to ask using the machine learning models was how heteromeric states compare to PPIs and expression in predicting CS scores. While this is addressed by the partial correlation analyses, it is not directly corroborated by the implemented models.

In the earlier version of the manuscript, we predicted the heteromeric state from the CS values, mRNA expression and the number of PPI partners. This allowed us to assess the predictive power of mRNA expression and number of PPI partners, however, it did not provide important information about the relative contribution of heteromeric state in the relationship. We thank the reviewer for pointing this issue out. In order to address this issue, we trained the machine learning models in the manner suggested by the reviewer i.e. we predict the deleteriousness from heteromeric state, mRNA expression and number of PPI partners.

In the revised manuscript we replace the previous indirect approach with the new direct approach.

Fig 4: Relationship between the effect of LOF of a gene on cell proliferation, mRNA expression and number of protein-protein interaction partners.

E) Feature importance (shown on the y axis) of the three factors as determined through four different classification models (shown on the x axis). Mean and standard deviation of the ROC AUC values across cross validations and bootstrapping runs (see methods) is plotted for each of the 4 classifiers. The CS values used for this analysis are mean of the CS values across all the CS

datasets. For similar analysis with the 4 individual CS datasets, see Appendix Fig S9 panel A to D.

In addition, we also use multiple linear regression (MLR) to determine the CS value from heteromeric state, mRNA expression and number of PPI interaction.

Appendix Fig S9: Feature importances of heteromeric state of the paralog, mRNA expression and number of PPI partners.

Multiple regression analysis to predict the deleteriousness of the paralog (CS value) from feature set consisting of mRNA expression and number of PPI partners. Inclusion of heteromeric status of the paralogs in the feature set improves the regression (estimated in terms of Pearson's correlation coefficient, r_p) indicating that heteromeric status of the paralog is one of the predictors of the deleteriousness of paralogs, albeit weaker one as compared to mRNA expression and the number of PPI partners. Additionally, inclusion of interdependence in the regression (interactions of degree 2) also improves the strength of regression, indicating the interdependence between the features is of important role. Shown in panels **E** and **F** is the analysis with heteromers defined by all and direct PPIs respectively. The results of multiple linear regression are similar in the two PPI datasets used.

Results of both the analyses corroborate the partial correlation analysis. We find that heteromeric state is indeed a predictor of deleteriousness of a gene albeit a weaker one compared to mRNA expression and number of PPI partners. In addition, MLR also allowed us to demonstrate the importance of the interdependence between features in determining the deleteriousness upon LOF.

Overall, we are very grateful for the reviewer's comment. The suggestions provided by the reviewer have greatly benefited the manuscript.

3. Page 15, line 6 : "In the case of all the CS datasets, at lower expression values, the difference in CS between paralogs and singletons is non-significant (Fig EV3E). The reason for this could be attributed to the small counts of mRNA expression generally being relatively more noisy". While expression measurements are indeed noisier at

lower expression levels, it is not immediately clear how this would explain the similar CS values between paralogs and singletons. Specifically, because the values of these two variables (paralog vs. singleton, CS value) are calculated without using any expression measurements.

Our rationale behind the given attribution was that more noisy expression values in general would make the expression levels unreliable. Thus they could possibly be of similar levels. Other than that, the explanation for the similar CS values could have been that the low expression genes have less difference in their deleteriousness upon LOF, arguably because they are the least essential.

Therefore, we revised the relevant sentence as follows (on page 10 line 15 to 18).

The reason for this could be attributed to the small counts of mRNA expression generally being relatively more noisy as well as lower fitness effects in general, making differences more difficult to detect.

We thank the reviewer for this point.

Minor points:

2. The order of labels in figure 2E could be re-arranged to reflect taxonomic hierarchy. In the figure legend (page 11, line 7) it says that paralogs are ordered by their age, but shouldn't then *Homo sapiens* be either at the top or bottom of the list?

The order of labels in Figure 2E is now rearranged in increasing order of their age, as suggested by the reviewer. This was indeed needed. We thank reviewer for this suggestion.

Fig 2. The LOF of paralogs that form heteromers is more deleterious than the LOF of non-heteromers.

E) Paralogs that form heteromers tend to be more deleterious upon LOF than other paralogs. Data from CS2.1 is shown, largely independent of the age of the paralog. In the legends, paralogs are ordered by their age. The CS values per class of paralogs (heteromer or not) and their age group are aggregated by taking

median across cell lines. Note that while heteromers are more deleterious in most of the age groups, in the case of 2 out of 10 age groups a reverse trend is observed. Distributions of the CS values per class of paralogs (heteromer or not) and their age group for this analysis are shown in Appendix Fig S5A. Similar analysis with dataset CS2 and for heteromers detected with 'direct PPI's only is shown in Appendix Fig S5 panels B to D.

3. According to the authors' reply, the X-axis of the left panel of figure 3 should say "CS mean (all paralogs)" instead of "CS mean (heteromer)". Also, it may be easier to read "proportion of heteromers" instead of "# of heteromers/ # of paralogs", in the figure.

There was an unfortunate error in the first sentence of the Figure 3 legend (“(x-axis)” was unnecessary), that might have lead the reviewer to suggest this. In fact, the left panel is correctly labeled as “CS mean (heteromer)”.

In order to avoid such confusion, we have now corrected the error in the first sentence of the Figure 3 legend and explicitly stated in the legends (page 25 line 7 to 8) that

In the left panel, average CS value of heteromers per category is shown on the x-axis.

With regards to reviewer's second suggestion, we have replaced "# of heteromers/ # of paralogs" with "proportion of heteromers" in the relevant figures (Fig 3, Fig EV2 and Appendix Fig S6). This change indeed improves the readability of these figures.

Fig 3. Association between the molecular functions of paralogs, their probability of heteromerization and the effect of gene LOF on cell proliferation.

Average CS values of paralogs (heteromer or not heteromer) belonging to a gene set were used in the analysis. On y-axis, GO molecular functions are sorted according to their proportion of heteromeric paralogs (i.e. # of heteromers/ # of paralogs, heteromers defined by ‘all PPI’). The size of the circles represent the number of paralog pairs in a category and the colors represent the proportion of heteromers in that category. In the left panel, average CS value of heteromers per category is shown on the x-axis. In the right panel, the difference between the average CS value of the heteromers and average CS value of the non-heteromers is plotted. The terms with significant difference between the average CS value of the heteromers and average CS value of the non-heteromers (estimated by two-sided t-test) are annotated with the blue edges. Descriptions of the representative significant GO terms with the highest difference are shown in the right side-panel. Spearman rank correlation between the proportion of the heteromers in the GO terms and the average CS value of paralogs in the term (r_s (# of heteromers / # of paralogs per term, CS mean of paralogs per term)) is shown in left right corner. Only GO molecular functions with more than 10% of the number of paralogs in all the gene sets are shown.

Similar analysis for the GO biological process and GO cellular component aspect, for the ‘all PPI’ based data are shown in Fig EV2. Similar analysis with the ‘direct PPI’ data is shown in Appendix Fig S6. See Dataset EV5 for GO terms and annotations

shown on this figure. Note that not all gene sets are independent because some genes are in several categories.

Accepted

28th August 2019

Thank you again for sending us your revised manuscript. We are now satisfied with the modifications made and I am pleased to inform you that your paper has been accepted for publication.

Corresponding Author Name: Christian R Landry
 Journal Submitted to: Molecular Systems Biology
 Manuscript Number: MSB-19-8871R